



# 1  Carbon and air pollutant emissions from China's cement industry
# 2  1990-2015: trends, evolution of technologies and drivers

Jun Liu[1, *], Dan Tong[1, *], Yixuan Zheng[1], Jing Cheng[1], Xinying Qin[2], Qinren Shi[2], Liu Yan[1], Yu Lei[3],
Qiang Zhang[1]
[1]Ministry of Education Key Laboratory for Earth System Modelling, Department of Earth System Science,
Tsinghua University, Beijing 100084, People's Republic of China
[2]State Key Joint Laboratory of Environment Simulation and Pollution Control, School of Environment, Tsinghua
University, Beijing 100084, People's Republic of China
[3]Chinese Academy for Environmental Planning, Beijing 100012, People's Republic of China
[*]These authors contributed equally to this work.
*Correspondence to*: Qiang Zhang (qiangzhang@tsinghua.edu.cn)
**Abstract.** China is the largest cement producer and consumer in the world. Cement manufacturing is highly energy-intensive,
and is one of the major contributors to carbon dioxide ($CO_2$) and air pollutant emissions, which threatens climate mitigation
and air quality improvement. In this study, we investigated the decadal changes of carbon dioxide and air pollutant emissions
for the period of 1990-2015, based on intensive unit-based information on activity rates, production capacity, operation status,
and control technologies, which improved the accuracy of the cement emissions in China. We found that, from 1990 to 2015,
accompanied by a 10.9-fold increase in cement production, $CO_2$, $SO_2$, and $NO_x$ emissions from China's cement industry
increased by 626%, 59%, and 658%, whereas CO, $PM_{2.5}$ and $PM_{10}$ emissions decreased by 9%, 66%, and 63%, respectively.
In the 1990s, driven by the rapid growth of cement production, $CO_2$ and air pollutant emissions increased constantly. Then,
the production technology innovation of replacing traditional shaft kilns with the new precalciner kilns in the 2000s markedly
reduced $SO_2$, CO and PM emissions from the cement industry. Since 2010, the growing trend of emissions has been further
curbed by a combination of measures, including promoting large-scale precalciner production lines and phasing out small ones,
upgrading emission standards, installing low-$NO_x$ burners (LNB) and selective noncatalytic reduction (SNCR) to reduce $NO_x$
emissions, as well as adopting more advanced particulate matter control technologies. Our study highlights the effectiveness
of advanced technologies on air pollutant emission control, however, $CO_2$ emissions from China's cement industry kept
growing throughout the period, posing challenges to future carbon emission mitigation in China.

## 27  1 Introduction

China is the largest cement producer and consumer in the world (Shen et al., 2015). As the basic industry for construction
materials, cement industry supports rapid social and economic development, but also suffers from high energy consumption
and serious air pollution problems. In 1990, China's cement output was 210 million tons (National Bureau of Statistics, 1991);



By 2015, the total cement production in China increased to 2360 million tons (National Bureau of Statistics, 2016), which was
10.2 times higher the output in 1990 and accounted for 58% of global total production in 2015 (USGS, 2015). The cement
industry is energy-intensive, representing 0.21 billion tons of coal consumption in 2012 and accounting for 6% of the total
industrial coal use (China Cement Association, 2015). It is a major $CO_2$ emitter due to high energy intensity and the dissociation
of carbonate during the clinker production process. At the same time, the cement industry contributes substantially to the
emissions of air pollutants, especially particles, $NO_x$, and $SO_2$. According to previous estimates for 2005, the cement industry
contributed 13%, 27%, 29%, 5%, 6% and 8% of national total $CO_2$, $PM_{2.5}$, $PM_{10}$, $SO_2$, $NO_x$, and CO emissions, respectively
(Lei et al., 2011a). The substantial emissions of $CO_2$ and air pollutants from China's cement industry poses challenges to global
climate mitigation and regional air quality improvements. Therefore, it is of great importance to develop a reliable and high-
resolution cement emission inventory to facilitate atmospheric chemistry modeling and support greenhouse gas mitigation and
air quality management.
Previously, greenhouse gas and air pollutant emissions from the cement industry in China were studied in several national and
regional inventories. The cement industry is the second largest anthropogenic contributor to $CO_2$ emissions, and many studies
focus on $CO_2$ emissions, energy intensity, energy-saving potential and the cost of the cement industry (Liu et al., 2013; Xu et
al., 2014; Shen et al., 2015; Zhang et al., 2015; Cai et al., 2016; Gao et al., 2017). In the atmospheric community, early studies
calculated cement air pollutant emissions based on the proportion of coal combusted in cement kilns (Streets et al., 2003;
Ohara et al., 2007). These studies did not distinguish the different kiln types and ignored process emissions, which resulted in
underestimations (Streets et al., 2006). The methodology was improved by introducing more detailed industrial source
categories, which allowed the distinction of combustion and process emissions (Zhang et al., 2006, 2007, 2009). Subsequently,
a dynamic and technology-based methodology with changing emission factors over a decade was developed, which provided
the historical trend of major air pollutants from China's cement industry (Lei et al., 2011a, 2011b). In addition to conventional
air pollutants, Hua et al. (2016) expanded the emission quantification to toxic heavy metals, including mercury, cadmium,
chromium, lead, zinc, arsenic, nickel and copper.
Despite remarkable improvements, there are still two major deficiencies in the current cement emission inventory of China.
First, owing to limited information available at the unit level, there is no cement emission inventory that estimates the
greenhouse gas and air pollutant emissions from individual clinker production lines and cement grinding plants, which is
insufficient to provide an accurate and high-resolution cement emission dataset for China. Second, with the economic
development and upgrade of emission standards, there has been a dynamic transition in cement production and emission control
technologies. Especially from 2010-2015, the production of cement has peaked, and the upgraded cement emission standards
(GB 4915-2013) promoted more advanced emission control technologies in the cement industry. These time-dependent
transitions should be implemented when constructing the historical trend of cement emissions in China.
Based on the background above, the aim of this study is to quantify the decadal changes of carbon dioxide and air pollutant
emissions from China's cement industry, investigate the evolution technologies, and identify the major drivers of the emission





trends over a period of 25 years. The analysis is based on intensive unit-based information on activity rates, production capacity,
operation status, and control technologies, which improves the accuracy of the estimation of cement emissions, provides a
comprehensive view of the effectiveness of technologies on air pollutant emission control in the past, and highlights the
challenges for future mitigation of carbon dioxide emissions in China.
**2 Materials and Methods**
**2.1 Activity rates**
In this study, we developed a unit- and technology-based methodology for $SO_2$, $NO_x$, CO, $CO_2$, $PM_{2.5}$, and $PM_{10}$ emissions in
the cement industry for the 1990-2015 period. We calculated only the direct emissions from cement production; indirect
emissions such as fuel use in the power plants due to electricity consumption and fuel use by vehicles for material transportation
were not included.
Cement production involves a series of complex processes, including three basic stages: raw material preparation, clinker
calcination and cement grinding (Cao et al., 2016). CO, $SO_2$, and $NO_x$ are only emitted from fuel combustion during the clinker
calcination process; thus, we estimated the emissions of these pollutants by the amount of coal consumed in the cement kilns,
and the coal use was calculated as the product of clinker production and annual energy intensity for the clinker production
process. $CO_2$ is primarily emitted from two sources: fuel combustion and calcination of calcium carbonates, which we treated
separately in the emission calculation. The emission of PM is more complex, involving the entire process of cement production,
including both organized and fugitive emissions. Following our previous study, we applied a similar model framework with a
dynamic methodology to consider the transition of various PM control technologies in different cement kilns under a series of
emission standards and control policies (Lei et al., 2011a, 2011b). The equations used to calculate various pollutants are
summarized in Table 1.
Detailed unit-level data from 2010-2015 were obtained from the China Ministry of Ecology and Environment (unpublished
data, hereafter referred to as the MEE database), including clinker and cement production, production capacity, operating and
retiring dates, PM and $NO_x$ control technologies, and the coordinates of each unit. Overall, the database consists of 3125
clinker production lines and 4549 cement grinding stations, of which 665 clinker production lines and 783 cement grinding
stations have been retired since 2010. Based on the MEE database for 2010-2015, we derived the activity rates for the period
1990-2009, with a combination of data from different sources. Provincial data on cement production during the 1990-2009
period were available in the China Statistical Yearbook (National Bureau of Statistics, 1991-2010), from which we calculated
the provincial clinker production based on the clinker-to-cement ratio collected from the China Cement Almanac (China
Cement Association, 2001-2015) and other literature (Xu et al., 2012, 2014; Gao et al., 2017). Then, we derived the unit-level
clinker and cement production for the years 1990-2009 by scaling the 2010 production of each unit to the corresponding years


according to its commission time. It should be noted that emission estimates prior to 2010 are more uncertain because
extrapolated parameters were used.
The energy efficiency of clinker production in China's cement industry has improved markedly over the past 25 years. The
average energy intensity of clinker production has decreased from 5.41 GJ/t-clinker in 1990 to 3.73 GJ/t-clinker in 2015
(National Bureau of Statistics, 2016). The historical energy intensities of different kiln types were not available from statistics,
but have been reported in several studies (Lei et al., 2011a; Xu et al., 2012; Shen et al., 2014; Zhang et al., 2015; Hua et al.,
2016). Originally, such information in a certain year was reported by the authority or research institutes, such as National
Development and Reform Commission and China Academy of Building Research, and then was interpolated between years
or averaged among different studies to derive the historical trend. There were discrepancies of the historical energy intensities
because the data sources and calculation methods were varied among different studies. For example, Lei et al (2011a) estimated
the average coal intensity of precalciner kilns in 1990 was 4.07 GJ/t-clinker, whereas 3.66 GJ/t-clinker from the estimation of
Xu et al (2012). To avoid the bias introduced by one particular study, we collected all the available data and generated a linear
regression between the logarithm of energy intensity (GJ/t-clinker) and time in years to predict the energy intensity in each
year (Fig.1), which enabled the calculation of coal consumption for each production line. According to the model regression,
the energy efficiency of precalciner kilns (PC) is distinctly higher than that of shaft kilns (SK) and the other rotary kilns (OR).
For example, the average energy intensity of PC, SK and OR kilns in 2010 was 3.39 MJ/t-clinker, 4.21 MJ/t-clinker and 4.84
MJ/t-clinker, respectively.
**2.2 Emission factors**
**2.2.1 $CO_2$**
$CO_2$ emissions originate from both the thermal decomposition of limestone and the burning of fuels in a cement kiln. The
methodology for estimating the $CO_2$ fuel emission factor follows the Intergovernmental Panel on Climate Change (IPCC)
Guidelines for National Greenhouse Gas Inventories (IPCC, 2006), as presented in Eq. 1.
$$EF_{coal,CO_2} = C \times R \times \frac{44}{12} \times H \qquad\qquad (1)$$
where $EF_{coal,CO2}$ refers to the fuel emission factor of $CO_2$ in g $kg^{-1}$, $C$ represents the carbon content of coal, $R$ is the oxidation
rate of coal, and $H$ refers to the heating value of coal. We adopted 25.8 kg $GJ^{-1}$, 98% and 20.908 GJ $kg^{-1}$ for the respective
values of $C$, $R$, and $H$ of the raw coal in China (Cui and Liu, 2008) and derived the $CO_2$ fuel emission factor as 1940 g $kg^{-1}$
coal (equivalent to 92800 kg $TJ^{-1}$ coal), which is consistent with the values of 92128~95700 kg $TJ^{-1}$ adopted in previous studies
(Xu et al., 2012; Hasanbeigi et al., 2013; Chen et al., 2015; Tan et al., 2016).
Process $CO_2$ emission is mainly from the decomposition of limestone, from calcium carbonate ($CaCO_3$) and magnesium
carbonate ($MgCO_3$) conversion to CaO and MgO. Therefore, the process $CO_2$ emission factor can be estimated by the
conservation of mass flow. In the absence of detailed data, it is widely accepted to use the IPCC default value of 510 kg $t^{-1}$



clinker, without considering the emissions from $MgCO_3$ (IPCC, 2006). The Cement Sustainability Initiative (CSI) suggested
calculating $CO_2$ emissions according to the CaO and MgO contents of clinker and recommended a default emission factor of
525 kg $CO_2$/t clinker (CSI, 2005). Recently, Shen et al. conducted a nation-wide sampling survey of 359 cement production
lines across 22 provinces of China and estimated the $CO_2$ emission factor with detailed chemical data and production
parameters, which was slightly lower than the values suggested by the international institutes (Shen et al., 2016). Therefore,
we adopted the process $CO_2$ emission factor from this local Chinese study, i.e., 519.66 kg/t-clinker, 499.83 kg/t-clinker, and
499.83 kg/t-clinker for PC, SK, and OR kilns, respectively.
**2.2.2 $SO_2$**
$SO_2$ is primarily emitted from coal combustion in kilns. After emission, a proportion of $SO_2$ is absorbed by the reaction with
calcium oxide (CaO). The $SO_2$ emission factor is estimated by a mass balance approach based on the sulfur content of coal
(Eq. 2):
$EF_{SO_2} = SCC \times (1 - S_r) \times (1 - A_r)$                 (2)
where $EF_{SO_2}$ represents the $SO_2$ emission factor, $SCC$ is the sulfur content of coal, $Sr$ is the faction of sulfur retention in ash,
and $Ar$ is the absorption rate of $SO_2$ as a result of reaction with calcium oxide in kilns.
The $SCC$ for each production line in each year was obtained from the provincial average $SCC$ compiled in our previous studies
(Lei et al., 2011a; Liu et al., 2015a) due to a lack of production-line-based data. The $SO_2$ absorption rate is approximately 70-
80% in PC kilns but is much lower in SK and OR kilns (Su et al., 1998; Liu, 2006). We assumed the $SO_2$ absorption rates for
PC, SK and OR to be 80%, 30%, and 30%, respectively (Lei et al., 2011a). The sulfur retention ratio in ash was assumed to be
25% for all the production lines. Because the calcination process can absorb a large proportion of $SO_2$ emissions, there are no
additional $SO_2$ abatement technologies in the cement industry. With the parameters above, the $SO_2$ emission from each clinker
production line was estimated as the product of coal consumption and the $SO_2$ emission factor (Table 1).
**2.2.3 CO**
CO is the incomplete combustion product of fuel use during clinker calcination in kilns and is highly dependent on temperature
and oxygen availability. Compared with rotary kilns, shaft kilns have a higher CO emission factor due to a lower operation
temperature and less oxygen availability. Based on local experiments, the CO emission factors from different types of kilns
were presented in previous studies on the emission inventory of China's cement industry (Lei et al., 2011a; Hua et al., 2016),
ranging from 12.9~17.8 kg/t-coal, 135.4~155.7 kg/t-coal, and 17.8 kg/t-coal for PC, SK, OR kilns, respectively. We
summarized these studies and adopted the median EFs from the literature for this study, as shown in Table 2.



### 2.2.4 NOx

Thermal NOx and fuel NOx are generated by fuel combustion in kilns during the clinker calcination process, with a high temperature exceeding 1400°C (Fan et al., 2014). Compared with shaft kilns, the operation temperature in rotary kilns is higher, which induces a higher NOx emission factor. In precalciner kilns, approximately half of the fuel is burnt in the preheater at a lower temperature, so the NOx emission factor is lower than that of other rotary kilns (Bo and Hu, 2010). Previously, NOx emission factors were presented in several Chinese local cement emission inventory studies (Wang et al., 2008; Lei et al., 2011a; Hua et al., 2016), ranging from 12.9~12.8 kg/t-coal, 1.2~1.7 kg/t-coal, and 13.6~18.5 kg/t-coal for PC, SK, and OR kilns, respectively. In addition, based on a nation-wide survey and measurements, the Chinese Research Academy of Environmental Sciences (CRAES) published the recommended NOx emission factor for the cement industry during China's first pollution census, i.e., the cement industry: 1.584~1.746 kg/t-clinker for precalciner kilns (equivalent to 9.7~10.7 kg/t-coal) and 0.202~0.243 kg/t-clinker for shaft kilns (equivalent to 1.0~1.2 kg/t-coal) (CRAES, 2011). By combining this research evidence, we adopted NOx emission factors for China's cement industry, as shown in Table 2.

Low-NOx burner (LNB) and selective noncatalytic reduction (SNCR) are the two major technologies to reduce NOx emissions from the cement industry. The application of LNB technology in China's cement industry dates back to the 1990s and has started to increase since 2009. During the 12th Five-Year Plan (FYP) period (2011-2015), the national emission of NOx was required to be cut by 10%. Driven by the policy requirements, newly established large kilns have been widely equipped with LNB devices, and a number of existing kilns have also been transformed to apply LNB technology. From 2011 to 2015, the proportion of kilns equipped with LNB technology increased from 3% to 40%, and the installation percentage of LNB in newly established kilns increased from 13% to 64%. The SNCR technology developed later in the 2000s. During the 12th FYP, the SNCR installation experienced unprecedented explosive growth. The penetration rate has increased even faster than that of the LNB technology, from 1% of all the kilns in service in 2011 to 88% in 2015.

However, the actual operation condition of the de-NOx facilities is less than satisfactory because the on-line NOx emission inspection system is not adequate in the cement industry. According to the MEE database, a large proportion of the de-NOx facilities (either LNB or SNCR) did not work properly after construction. For example, during the 2013-2015 period, there were ~800, ~1300 and ~1400 cement kilns equipped with SNCR systems, but only 51%, 54%, and 73% of these respective facilities were operating under normal conditions. Based on the information above, we assumed that the de-NOx devices were not in service before 2010, and the net NOx reduction rates from 2010-2015 for each production line were directly obtained from the MEE database.

### 2.2.5 PM

The particulate matter (PM) emissions are classified into three parts in this study: clinker production (including quarrying, crushing, calcination, and other processes), cement grinding, and fugitive emissions. The emission of PM is determined by the unabated emission factor of these processes and the reduction rates of PM emission control technologies. Since the PM





emission factors are clinker and cement output-based factors, we did not specifically distinguish the fuel emissions from
process emissions of PM in this study. We collected the unbated PM emission factor for clinker production and cement grinding
from previous Chinese local studies (Lei et al., 2011a; Hua et al., 2016) and the recommended value compiled by CRAES
during China's first pollution census (CRAES, 2011), from which we adopted the median value as the unabated PM emission
factors for this study (Table 3). The mass fractions of $PM_{2.5}$, $PM_{2.5-10}$, and $PM_{>10}$ relative to total particulate matter were derived
from our previous study (Lei et al., 2011a).
Due to limited information available, the fugitive PM emissions from the cement industry have not been elaborately studied
before. Tang et al (2018) calculated the organized and fugitive PM emissions from the cement-producing process and estimated
that the fugitive emissions contributed 44% of the total PM emissions in 2014 in China. Following the same methodology,
Wang et al (2018) estimated non-fugitive and fugitive PM, $PM_{10}$, and $PM_{2.5}$ emissions for the Beijing-Tianjin-Hebei region in
2016. The abated fugitive PM emission factors used in their study were 0.1~0.4 kg t$^{-1}$, 0.7 kg t$^{-1}$, and 0.6 kg t$^{-1}$ for PC, SK, and
OR kilns, respectively, and 0.2~0.3 kg t$^{-1}$ for the cement grinding process. However, these emission factors were not directly
applicable to establish the historical emission trend because the details on control efficiencies were missing. In this study, we
adopted the median values of unabated fugitive PM emission factors compiled by CRAES for China's first pollution census
(CRAES, 2011) and used the mass fraction of PM with different diameters from Wang et al (2018) to derive the size-specific
PM emission factors (Table 3). The size distributions of $PM_{2.5}$, $PM_{2.5-10}$, and $PM_{>10}$ in fugitive PM emissions were assumed to
be 10%, 20%, and 70% for all the fugitive emission processes (Wang et al., 2018).
There are five major types of PM removal technologies in China's cement industry, i.e., cyclone (CYC), wet scrubber (WET),
electrostatic precipitator (ESP), high-efficiency electrostatic precipitator (ESP2), and bag filters (BAG). We obtained the PM
removal technology application for each production line in 2010 from the MEE database and developed the technology
evolution model over the 1990-2015 period following our previous methodology (Lei et al., 2011a). Over the past decades,
China has progressively issued four editions of emission standards for air pollutants in the cement industry (GB 4915-1985,
GB 4915-1996, GB 4915-2004, and GB 4915-2013) and has successively strengthened the particulate matter concentration
limits of flue gas in kilns from 800 mg m$^{-3}$ to 20 mg m$^{-3}$. The fugitive PM emissions limits have also been included in the
standards since GB 4915-1996 (Table S1). According to the concentration limits of the four phases of emission standards, we
divided the entire study period into four phases, i.e., 1990-1996, 1997-2004, 2005-2013, and 2014-2015. In each phase, the
newly built units were designed to be equipped with the current advanced PM removal technologies recommended by the
documentation for the compilation of emission standards of air pollutants for the cement industry. For the existing units, we
combined the limited information on the penetration of PM control technologies from the MEE database and environmental
statistics and built an evolution model to perform the technology transformation for the in-fleet units step by step, assuming
that the larger and younger units were prioritized for technology upgrading and transformation. Finally, based on the removal
efficiencies of each technology (Lei et al., 2011a) listed in Table 4, we modeled the evolution of unit-based PM emission
factors for the 1990-2015 period (Fig. 2).





For fugitive PM emissions, there are a variety of control technologies, such as covering the open storage of materials, collecting
dust by PM removal facilities, reducing the transportation distance of raw materials, increasing the cleaning frequency of road
dust, and so on. However, information on the implementation details of these technologies was scarce, which hindered us from
establishing the unit-level technology evolution. Therefore, we estimated the average abatement rate of fugitive dust for the
entire cement industry. According to the on-site measurements conducted by the China Building Materials Academy in 2009,
the typical fugitive dust concentration observed 20 m from the factory boundary in the cement industry was 0.3368~2.56 mg
m$^{-3}$ (Wang et al., 2009). Therefore, we assumed the upper limit of 2.56 mg m$^{-3}$ as the unabated fugitive dust concentration,
estimated the average fugitive PM abatement rates for each phase of emission standards, and interpolated the abatement rates
across the entire study period (Fig. S1).
**2.3 Uncertainty analysis**
Following the methodology demonstrated in our previous studies on the power sector (Liu et al., 2015a; Tong et al., 2018), we
performed an uncertainty analysis of the emissions estimated in this study at the national and unit levels with a Monte Carlo
approach. The "uncertainty" was estimated by the 95% confidential interval (CI) around the central estimate of the emission
from 10000 Monte Carlo simulations with a specific probability distribution of input parameters, such as activity rates, coal
intensity, emission factors, abatement efficiency of control technologies, and so on. The probability distributions of the related
parameters were based on adequate measurements (e.g., $CO_2$ emission factors), model regressions (e.g., coal intensity), a
literature review (Lu et al., 2011; Zhao et al., 2011; Liu et al., 2015a; Wang et al., 2019), and our own judgment. Table S2
presents the detailed information on the probability distribution of the parameters used in the uncertainty analysis.
For the unit-level uncertainty analysis, the uncertainty level of emission estimates in the 1990-2009 period was regarded as
larger than that in the 2010-2015 period because all the unit-level data were directly available from the MEE database for the
later period. The uncertainties conveyed by input parameters such as activity rates, emission factors, and control technologies
could vary with time. Therefore, we also estimated the uncertainty ranges of one representative clinker production line (a
precalciner kiln with a capacity of 4000 t-clinker/day, equipped with LNB, SNCR, and a bag filter in 2015) for 2000 and 2015
to demonstrate the change in unit-level uncertainties. The probability distribution of the parameters that are different from the
parameters used in the national uncertainty analysis is listed in Table S3.
**3 Results**
**3.1 Historical cement production and evolution of technologies**
Driven by the economic development and urbanization process, China has experienced rapid growth in cement production and
technology evolution in the cement industry. From 1990 to 2014, the production of cement and clinker increased from 0.21
and 0.16 billion tons to 2.5 and 1.4 billion tons, i.e., by 10.9 and 8.2 times, respectively (Fig. 3 and Table 5). The total



production started to diminish in 2015 as a consequence of recent clean air actions (Zheng et al., 2018). Cement is a blending
mixture of clinker and other additives, such as coal fly ash, plaster, clay, and so on. Typically, replacing clinker with other
additives can reduce the energy intensity and $CO_2$ emissions. With raised clinker quality from an increased number of new
kilns, less clinker is required to produce a given strength of cement; thus, the clinker-to-cement ratio decreased from 74% in
1990 to 57% in 2015.
In China, the shaft kilns, precalciner kilns and other rotary kilns are the major kiln types for clinker calcination, representing
68%, 7%, and 25%, respectively, of the total clinker production in 1990. Prior to 2004, shaft kilns dominated China's cement
industry, accounting for over half of the clinker production; they were gradually replaced by new precalciner kilns from 2005
to 2015. Currently, the precalciner kiln is the dominant kiln type in China, and the proportions of the other two types are
negligible. In accordance with the transition of kiln types, the share of kilns with different designed capacities also varied
during the 1990-2015 period. The small-scale production lines (<2000 t-clinker/day), contributed mostly by shaft kilns, had a
dominating role in the 1990-2000 period, with a proportion exceeding 85%, whereas the share of large-scale production lines
(≥2000 t-clinker/day), majorly contributed by precalciner kilns, increased sharply afterwards, from 14% in 2000 to 97.5% in

261    2015.

To fulfill the rapidly growing demand for cement products and to achieve ever-stringent clean air targets at the same time,
China's cement industry has undergone dramatic transitions in the production technology of cement kilns in recent years since
2010. Fig. 4 shows the share of different kiln types in the newly built and retired production lines and the cumulative ratio of
newly built and retired production lines by unit capacity. During the 2010-2015 period, there were 688 newly built cement
production lines, of which the precalciner kilns shared a dominant proportion of 95%. In contrast, there were 665 retired
cement production lines, of which the shaft kilns had a majority proportion of 79%. In response to the energy conservation
and emission reduction policies, the number of newly built production lines decreased, and the capacity of these newly built
production lines increased year by year. On the other hand, the number of retired production lines reached a peak during 2012-
2013, and the capacity retirement dramatically extended to the large-scale production lines during 2014-2015, likely driven by
the implementation of the new emission standard of the cement industry (GB4915-2013) and the Clean Air Action Plan issued
in 2013.
**3.2 Emission trends**
Table 6 and Fig. 5 summarize the historical emissions of gaseous species and particulate matter in China's cement industry
from 1990 to 2015. During the 25 years, the cement production increased dramatically, by 10.5 times. During that time, the
$CO_2$, $SO_2$, and $NO_x$ emissions from the cement industry increased by 627%, 56%, and 659%, whereas the CO, $PM_{2.5}$ and $PM_{10}$
emissions decreased by 9%, 63%, and 59%, respectively, indicating that significant technology transitions occurred in the past
25 years. As a major air pollution source in China, the cement industry contributed approximately 4%, 7%, 2%, 9%, 11%, and





10% of the national anthropogenic $SO_2$, $NO_x$, CO, $PM_{2.5}$, $PM_{10}$, and $CO_2$ emissions (emissions from other sources were
estimated by MEIC model), respectively, in 2015.

### 3.2.1 $CO_2$ emissions

Fig.6 shows the historical $CO_2$ process and fuel emissions in China's cement industry. The total emissions of $CO_2$ increased
in line with the growth of cement production. Driven by the 8.2-fold increase in clinker production, the $CO_2$ emissions in
China's cement industry increased from 0.15 Pg in 1990 to 1.18 Pg in 2014, i.e., by 6.8 times (Fig. 5). The growth of $CO_2$
emissions was slightly lower than that of clinker production due to the offset effect from improved energy efficiency. From
1990 to 2015, the $CO_2$ process emissions increased from 77.7 Tg to 694.2 Tg, i.e., by 7.9 times, which was consistent with the
growth of clinker production, whereas the $CO_2$ fuel emissions increased more slowly, from 73.5 Tg to 405.9 Tg, i.e., by 4.5
times, because the energy intensity of cement kilns decreased significantly at the same time (Fig. 6). During the 1990-2015
period, the energy intensity of precalciner kilns, shaft kilns and the other rotary kilns decreased by 17%, 16% and 27%,
respectively. As a result, the proportion of $CO_2$ emissions from coal consumption also decreased from 49% in 1990 to 37% in
2015. By 2015, cement and clinker production decreased, and the corresponding $CO_2$ emissions dropped to 1.10 Pg.

### 3.2.2 Gaseous air pollutant emissions

Fig. 7 presents the historical emissions of gaseous air pollutants, including $SO_2$, CO, and $NO_x$, by different kiln types from
1990 to 2015. During the 1990-2003 period, the $SO_2$ emissions increased from 0.43 Tg to 1.46 Tg, at an annual increasing rate
of 10%, driven by the growth of cement production, which was mainly manufactured in the highly polluting shaft kilns (Fig.
7). Then, the $SO_2$ emissions decoupled with the increasing trend of cement production and decreased to 0.66 Tg in 2015. The
emission decrease was due to the expanding technology transition from the old and polluting shaft kilns to the new and cleaner
precalciner kilns, which resulted in a much lower $SO_2$ emission factor (Table 2). The CO emissions had a similar trend as the
$SO_2$ emissions.
In contrast, the $NO_x$ emissions exhibited a longer period of growth than other gaseous pollutants. In the 1990s, the $NO_x$
emission gradually increased at an annual growth rate of 6.9% with the increase in cement production, which was mainly
manufactured in the shaft kilns and other rotary kilns. Since 2003, the rapid growth of cement production and the wide
promotion of precalciner kilns to substitute the shaft kilns have accelerated the growth of $NO_x$ emissions from the cement
industry because the precalciner kilns have a higher $NO_x$ emission factor under a higher operation temperature (Table 2). As
a result, the $NO_x$ emissions increased sharply from 0.64 Tg in 2003 to 2.13 Tg in 2012, i.e., by 234%. During the 2011-2015
period, the 12[th] FYP required a national target of reducing $NO_x$ emissions by 10%, which promoted the wide installation of
LNB and SNCR devices in the cement industry (Fig. 8). In 2011, only 11% and 1% of the clinker was manufactured in kilns
equipped with LNB and SNCR facilities, whereas by 2015, the percentages sharply increased to 50% and 97%. However, the
actual operation condition of the de-$NO_x$ facilities was far from satisfactory. In 2011, among all cement kilns equipped with



LNB or SNCR devices, only 20% of the clinkers were produced under normal operating conditions of DeNO$_x$ devices, and in
2015, the percentage increased to 82%. Meanwhile, with technology improvements and a wider application of the DeNO$_x$
technologies, the national average NO$_x$ removal efficiency increased during the 5-year period and remained relatively stable
at 32%-43%.

### 3.2.3 Particulate matter emissions

Fig. 9 depicts the PM$_{2.5}$ and PM$_{10}$ emissions by different processes, including clinker calcination (precalciner kilns, shaft kilns
and the rotary kilns), cement grinding and fugitive emissions. The respective PM$_{2.5}$ and PM$_{10}$ emissions decreased from 2.11
Pg and 3.32 Pg in 1990 to 0.77 Pg and 1.37 Pg in 2015, with two peaks occurring in 1996 and 2003, due to the combined
effects of cement demand growth and environmental policies. The estimated PM emission trend from 1990-2008 was
consistent with that reported in our previous study (Lei et al., 2011a). From 1990 to 1995, PM emissions increased rapidly,
driven by the growth of cement production. The decline of PM emissions after 1996 was due to the implementation of the new
emission standards for the cement industry issued in 1996 (GB4915-1996, Table S1) and the slowing down of the economy in
the Asian financial crisis. The PM emissions rebounded after the financial crisis but dropped again after 2003, despite a
continuous increase in cement production at an annual growth rate higher than 10%. The decline of PM emissions after 2003
was due to the nation-wide replacement of the shaft kilns with precalciner kilns and the application of high removal efficiency
PM control technologies, such as high-efficiency ESP and bag filters. During the 2003-2015 period, the Chinese government
successively issued two versions of the air pollutant emission standard for the cement industry (GB4915-2004, GB4915-2013),
which promoted the technology transition of cement production and PM control in China's cement industry.
The contribution from different processes to the total PM emissions changed significantly during the 25 years. In 1990, the
polluting shaft kilns had the largest contribution to PM emissions, followed by other rotary kilns and the cement grinding
process. In 2015, the emission from the precalciner kilns was the largest contributor, followed by fugitive emissions and cement
grinding processes. The PM emissions from rotary kilns and shaft kilns in 2015 were negligible. Over the whole study period,
the contribution of organized emissions from clinker calcination and the cement grinding process was sharply reduced by the
implementation of improved PM control technologies, whereas the contribution of unorganized fugitive emission gradually
occupied a larger proportion, from 2% to 17% for PM$_{10}$ and from 1% to 13% for PM$_{2.5}$, indicating the necessity of more policy
arrangements targeting fugitive emissions in China's cement industry.
Fig. 10A further shows the historical PM$_{2.5}$ emissions from the clinker calcination process by production capacity. Prior to
2003, the small-scale capacities (<2000 t-clinker/day) dominated the emissions of China's cement industry, with a contribution
of 89%, due to their leading roles in clinker production (Fig. 3) and the inefficiency of PM control technologies. After 2003,
driven by the rapid development of new precalciner kilns, the share of small-scale production lines gradually declined (Fig. 3).
However, a considerable fraction of PM$_{2.5}$ emissions were still disproportionately produced by a small fraction of clinker
production. Fig. S2 presents the PM control technology penetration in production lines by different clinker production





capacities and the proportion of different capacities relative to the number of production lines, clinker production, and PM$_{2.5}$
emissions in 2010 and 2015. In 2010, the small production lines (<500 t-clinker/day) only represented 7% of the clinker
production but were responsible for 17% of the PM$_{2.5}$ emissions because more than 20% of the production lines were still
equipped with the outdated cyclone or wet scrubbers to reduce PM emissions (Fig. S2A). In 2013, the emission standard for
air pollutants was strengthened to fulfill the targets under the Clean Air Action Plan (GB 4915-2013), which accelerated the
phase-out of the small and outdated capacity and the transition of bag filters to meet the latest emission legislation. By 2015,
69% of the clinker was produced in the cement kilns with a capacity that exceeded 4000 t-clinker/day, and the overall
penetration rate of the bag filters reached 87% (Fig. S2B). Fig. 10B shows the changing routes of PM$_{2.5}$ emission distribution
in production lines sorted by clinker production capacity. Overall, during the 2010-2015 period, the contribution of small
capacities to the total PM$_{2.5}$ emissions decreased significantly, and the proportion of large capacities gradually increased as a
result of the rapid evolution of production technology in China's cement industry during recent years.

**3.3 Provincial distribution of emissions**

Fig. 11 shows the provincial distribution of the clinker production and emissions of CO$_2$, SO$_2$, CO, NO$_x$, and PM$_{2.5}$ from
China's cement industry in 2015. Anhui was the leading province with respect to CO$_2$ and air pollutant emissions due to its
prominent role in clinker production nationwide. In 2015, the clinker output in Anhui was 135 Tg, accounting for 9.5% of the
national total, whereas the cement output in Anhui was only 131 Tg (5.5%). The overall clinker to cement rate in Anhui was
1.03, while the national clinker to cement rate was only 0.57, indicating that Anhui exports a large amount of clinker to other
provinces (Liu et al., 2018; Shan et al., 2019). At the same time, it bears a heavier burden of emissions and air pollution from
the cement industry than other provinces. In addition to Anhui, Guangdong, Sichuan, Henan, Shandong, and Guangxi were
also important provinces for clinker production and emissions. The total emissions of the above six provinces contributed to
40%, 36%, 39%, and 38% of CO$_2$, PM$_{2.5}$, NO$_x$, and SO$_2$ emissions, respectively, driven by a 40% share of the national total
clinker production. In general, the provincial contribution of CO$_2$ emissions was consistent with the provincial clinker
production, but the provincial contribution of air pollutants was not always consistent. For example, Sichuan, Guizhou,
Guangxi, and Chongqing were the first four largest provinces with respect to SO$_2$ emissions, together contributing to 36% of
the national total, but they were not the first four leading provinces of clinker output because the sulfur content of coal in these
four provinces was much higher than that in other provinces. Regarding PM$_{2.5}$ and NO$_x$ emissions, the variation in the
penetration of end-of-pipe control technologies was another crucial factor in determining the differences in emissions. For
example, Yunnan was the sixth largest province with respect to NO$_x$ emissions, but with only moderate clinker output in 2015,
since the average NO$_x$ removal percentage achieved by LNB and SNCR devices was only 13% in Yunnan, much lower than
the national average of 30%.



## 4 Discussion

### 4.1 Uncertainty analysis

The uncertainties of the emission estimation in the study were quantified at both national and unit levels. We overlaid the uncertainty ranges of the national estimation in Fig. 12 and Fig. 13 with the emission estimates from various studies. Based on the 10000 Monte Carlo simulations, the average uncertainty ranges of the national estimates were -27 to 30%, -20 to 21%, -18 to 19%, -12 to 14%, -20 to 22%, and -16 to 17% for $SO_2$, $NO_x$, CO, $CO_2$, $PM_{2.5}$, and $PM_{10}$, respectively, in 2015. The uncertainties arising from clinker and cement production and coal consumption contributed to the uncertainties of all species. The uncertainty of $SO_2$ emissions was primarily contributed by the uncertainties from the sulfur content of coal, sulfur retention in ash, and the sulfur absorption rates of clinker during calcination, whereas the sources of the uncertainties for $NO_x$ and PM emissions were dominated by uncertainties in the unabated emission factors and the removal efficiency of technologies. During 1990 and 2015, the respective uncertainty ranges of $SO_2$, $NO_x$, CO, $CO_2$, $PM_{2.5}$, and $PM_{10}$ emissions had significantly decreased (Fig. 12 and Fig. 13), denoting the accuracy improvements from the input data. During the 2010-2015 period, the unit-level information on activity and control technologies was directly obtained from the MEE database, whereas for the past years, extrapolations and assumptions were made on the transition of activities, emission factors, technology penetration and efficiencies, which resulted in higher uncertainties. In particular, for the $PM_{2.5}$ and $PM_{10}$ emissions, the uncertainty ranges shrunk significantly after 2010, since the wide application of high-efficiency bag filters with lower uncertainty was assumed to effectively reduce the rise of PM emissions, and the increase of fugitive emissions were much lower than the decrease of other process emissions. Our estimation of the uncertainty ranges was comparable with the recent united-based emission inventory of China's power plants (Liu et al., 2015a) and the iron and steel industry (Wang et al., 2019) but was significantly narrower compared with previous studies relying only on statistics (Zhao et al., 2011, 2017).

We further quantified the uncertainty ranges of emission estimation at the unit level. For the selected production line (a precalciner kiln with a capacity of 4000 t-clinker/day, equipped with LNB, SNCR, and bag filters in 2015), the uncertainty ranges declined significantly from -34-42%, -30-29%, -25-29%, -21-22%, -37-51%, and -35-45% in 2000 to -29-31%, -21-24%, -19-21%, -12-13%, -35-40%, and -28-31% in 2015 for $SO_2$, $NO_x$, CO, $CO_2$, $PM_{2.5}$, and $PM_{10}$ emissions, respectively, showing consistent trends with the national uncertainty ranges. At the same time, the unit-specific uncertainty ranges were slightly broader than the national estimates because parts of the national uncertainties could be offset during the unit-level summation calculations.

### 4.2 Comparison with previous studies

We compared our estimates of $CO_2$, $SO_2$, $NO_x$, CO, $PM_{2.5}$, and $PM_{10}$ emissions with other bottom-up emission inventories (Lei et al., 2011a; Ke et al., 2012; Ministry of Ecology and Environment of the People's Republic of China, 2012; Crippa et al., 2014; Xu et al., 2014; Liu et al., 2015b; Zhang et al., 2015; Cai et al., 2016; Hua et al., 2016; Gao et al., 2017; Jiang et al., 2018; Shan et al., 2019), as shown in Fig. 12 and Fig. 13. There is abundant literature on $CO_2$ emissions (Fig. 12). Direct $CO_2$



emissions include both process emissions from the decomposition of limestone and fuel emissions from the burning of coal.
Basically, our estimates of total direct $CO_2$ emissions had a consistent trend with other studies (Fig. 12C), and the variations
among different studies mainly originated from the variations in the estimates of $CO_2$ fuel emissions. The $CO_2$ process
emissions were directly calculated as the product of clinker output and the process $CO_2$ emission factor, which was highly
consistent among different studies (Fig. 12A). However, there were larger discrepancies in the estimates of $CO_2$ fuel emissions
because the amount of coal use in China's cement industry was not directly available in the statistics and was derived through
the coal intensity value, which resulted in higher variations than the estimates of process emissions (Fig. 12B). Therefore,
several studies, such as Liu et al., (2015b) and EDGAR v4.3 (Crippa et al., 2014), only reported the estimates for $CO_2$ process
emissions and did not separate the $CO_2$ fuel emissions of the cement industry from the total industrial $CO_2$ fuel emissions. In
Fig. 12B, the lower estimates of $CO_2$ fuel emissions presented by Shan et al., (2019) were due to the application of a lower
$CO_2$ fuel emission factor (499 g $CO_2$ kg$^{-1}$ coal vs. 1940 g $CO_2$ kg$^{-1}$ coal in this study), whereas the higher estimates of $CO_2$
fuel emissions reported by Zhang et al., (2015) were likely due to the application of a higher $CO_2$ fuel emission factor.
As shown in Fig. 13, for $SO_2$ emissions, our study presented consistent trajectories with two other Chinese studies (Hua et al.,
2016; Lei et al., 2011a), whereas for CO emissions, the estimates by Hua et al., (2016) were slightly lower than the lower
boundary of the 95% CI calculated in this study after 2009, which was likely due to the adoption of lower energy intensity in
clinker production by Hua et al., (2016). For $NO_x$ emissions, all studies exhibited a similar growth trend before 2010 (Lei et
al., 2011a; Hua et al., 2016) and characterized a consistent declining trend from 2011-2015 (Ministry of Ecology and
Environment of the People's Republic of China, 2012; Jiang et al., 2018), but Lei et al., (2011a) had slightly higher estimates
of $NO_x$ emissions than the higher boundary of the 95% CI of this study due to the selection of higher $NO_x$ emission factors.
For PM emissions, all the studies indicated a similar trend during the 25 years, with two peaks occurring in the 1990s and
2000s. Even though we separately considered cement grinding and fugitive emission processes, in general the $PM_{2.5}$ and $PM_{10}$
emission estimates by the two other studies (Lei et al., 2011a; Hua et al., 2016) lay within the uncertainty ranges of this study,
since the other two studies also included the grinding process in the total PM emission factors, and the fugitive emissions were
much lower than the emissions from clinker calcination process. In fact, the central estimates of this study were significantly
lower than those in the previous studies because we integrated the recent Chinese local measurements of PM emission factors
in clinker calcination process obtained during China's first pollution census (CRAES, 2011), which were lower than those in
the previous studies [129 g/kg in this study vs. 168 g/kg reported by Lei et al. (2011a) for SK kilns]. In addition, we estimated
a more rapid declining trend of PM after 2009, which differs from the relatively stable trend presented by Hua et al. (2016),
likely because these authors failed to characterize the PM emission control progress in China's cement industry in recent years.
**5 Conclusions**
This study estimates the trends of carbon dioxide and air pollutant emissions in China's cement industry during 1990-2015
and investigated the drivers behind the trends, with a combination of unit-based information on activities, control technologies,





building and retiring dates for ~3100 clinker production lines and ~4500 cement grinding stations. According to our estimates,
$SO_2$, $NO_x$, CO, $PM_{2.5}$, $PM_{10}$ and $CO_2$ emissions in China's cement industry were 0.66 Tg, 1.59 Tg, 3.46 Tg, 0.77 Tg, 1.37 Tg,
and 1.10 Pg, respectively, in 2015. From 1990 to 2015, the $CO_2$, $SO_2$, and $NO_x$ emissions from the cement industry increased
by 627%, 56%, and 659%, whereas the CO, $PM_{2.5}$ and $PM_{10}$ emissions decreased by 9%, 63%, and 59%, respectively.
Significant technology transition has occurred in the past 25 years, resulting in different emission trajectories of different
species. The $CO_2$ emissions experienced an overall growth driven by the rapid growth of cement production, whereas the $SO_2$
and CO emissions declined since 2003 with rapid technology transition from the old shaft kilns to the new precalciner kilns,
while the end-of-pipe emission control measures were the major reasons for the decline in the PM and $NO_x$ emissions.
In the recent years of 2010 to 2015, significant changes have occurred in China's cement industry, driven by the growing
demand for cement products and offset by the strengthened emission control policies. Numerous precalciner kilns with a
capacity greater than 4000 t-clinker/day were built to replace the outdated small shaft kilns. The end-of-pipe emission control
facilities, such as LNB, SNCR and bag filters, were widely promoted to reach the new emission standard (GB4915-2013) of
400 mg m$^{-3}$ for $NO_x$ and of 30 mg m$^{-3}$ for particulates since 2014. Meanwhile, for the first time, cement production peaked in
2014. The respective penetration rates of LNB and SNCR increased from 11% and 1% in 2011 to 50% and 97% in 2015, which
constrained the rapidly growing trend of $NO_x$ emissions. Before 2003, the small capacities (<2000 t-clinker/day) contributed
to over 75% of the clinker output, then the share of large-scale production lines (≥2000 t-clinker/day), majorly contributed by
precalciner kilns, increased sharply afterwards. Since the precalciner kilns have lower emission factors of $SO_2$ and CO, and
higher penetration of high-efficiency PM and $NO_x$ removal technologies, the elimination of small capacities achieved
substantial emission reductions in the cement industry. Besides, though not involved in this study due to data unavailability,
large-scale production lines have higher energy efficiencies than the small capacities, which contribute to additional reductions
of $CO_2$ and air pollutant emissions. Great emission reduction potentials can be achieved in the cement industry in the near
future by eliminating the excess and outdated capacities, strengthening the on-line emission monitoring systems and promoting
ultralow emission technologies.
This study has several uncertainties and limitations. The emission estimates for the 1990s and 2000s were considered to have
higher uncertainties than the estimates for the years of the 2010s due to incomplete unit-level information for the early years.
More unit-based data for the past years need to be collected from provincial and subprovincial departments to improve the
temporal coverage. This study does not consider the application of wastes as fuels in the cement industry. In 2017, there were
around 100 cement kilns that can burn household wastes, municipal sludge, and hazard wastes as substitutes for coal use, but
the overall thermal substitution ratio was only 1.5%, due to limited waste disposal rates in the kilns and the low calorific value
of waste fuels (Gao, 2018). We thus did not take into account the use of waste-derived fuels in the study. Compared with the
$CO_2$ emission factors, local measurements for the emission factors of air pollutants are still limited. More on-site measurements
are needed to better characterize the source-specific emission factors and particle-size distributions to improve the
understanding of emissions from China's cement industry.



**Data availability**

Data generated from this study are available from the corresponding author upon request (qiangzhang@tsinghua.edu.cn). Unit-level data used in this study are owned and managed by the Ministry of Ecology and Environment, which are confidential and not available to the public.

**Author contributions**

Q.Z. designed the study; J.L. and D.T. calculated emissions; Y.Z., J.C., X.Q., Q.S., and Y.L. helped on data processing; Q.Z., J.L., D.T., and Y.L. interpreted the data; J.L. and D.T. prepared the manuscript with contributions from all co-authors.

**Competing interests**

The authors declare that they have no conflict of interest.

**Acknowledgements**

This work was supported by the National Natural Science Foundation of China (91744310 and 41625020), Beijing Natural Science Foundation (8192024), and China Postdoctoral Science Foundation (2018M641382). We thank Youwang Deng for collecting data at the early stage of this work.

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





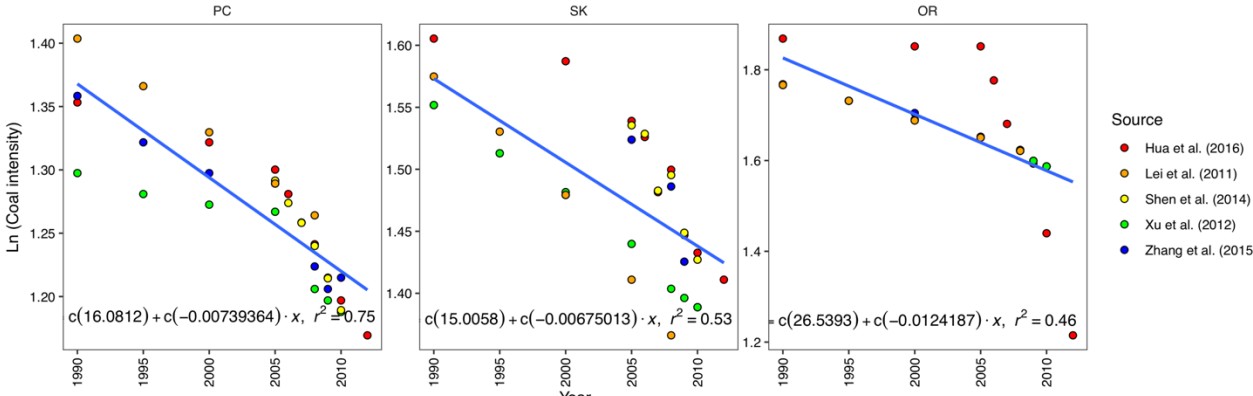

Figure 1: Linear regression of the logarithm of coal use intensity for different kiln types. The kiln types include precalciner kilns (PC), shaft kilns (SK) and the other rotary kilns (OR).





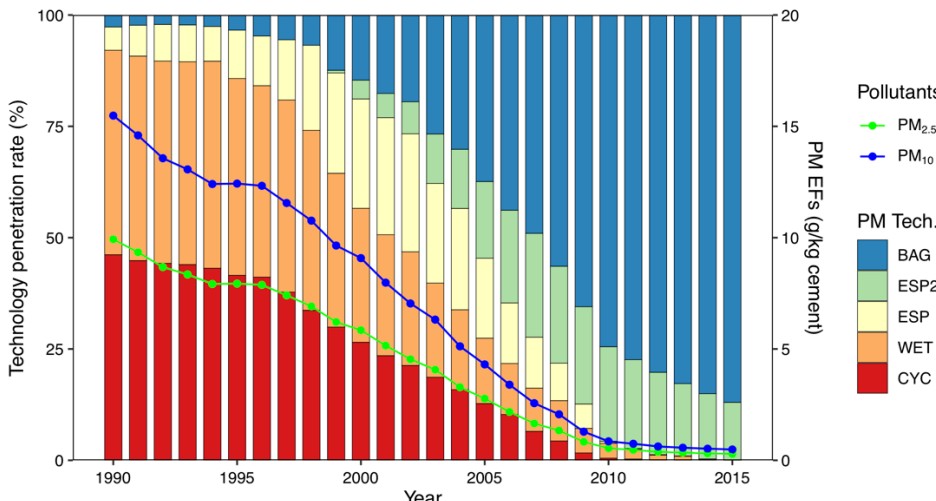


**Figure 2: Evolution of PM$_{2.5}$ removal technology and the average PM emission factors for each year.**



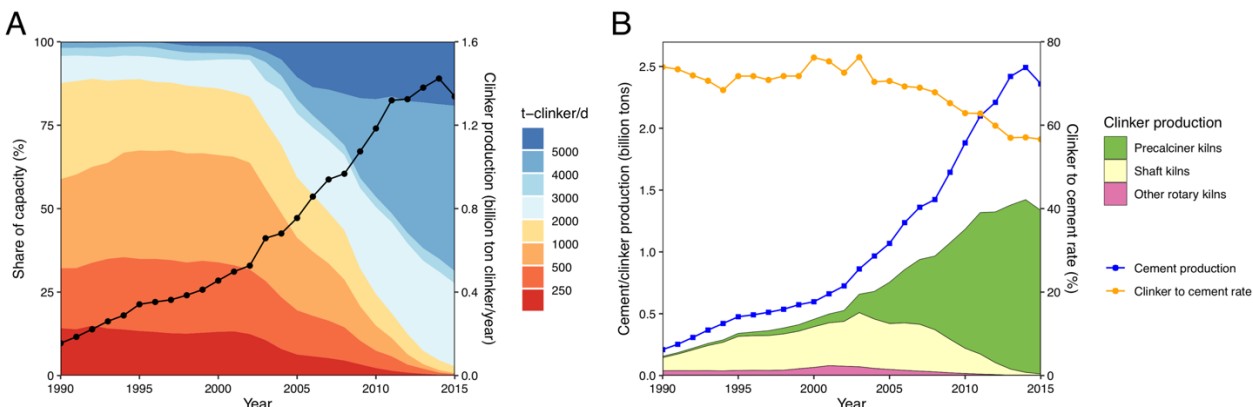


**Figure 3: Clinker production by designed capacity (t-clinker/day) (A) and by different kiln types (B).**




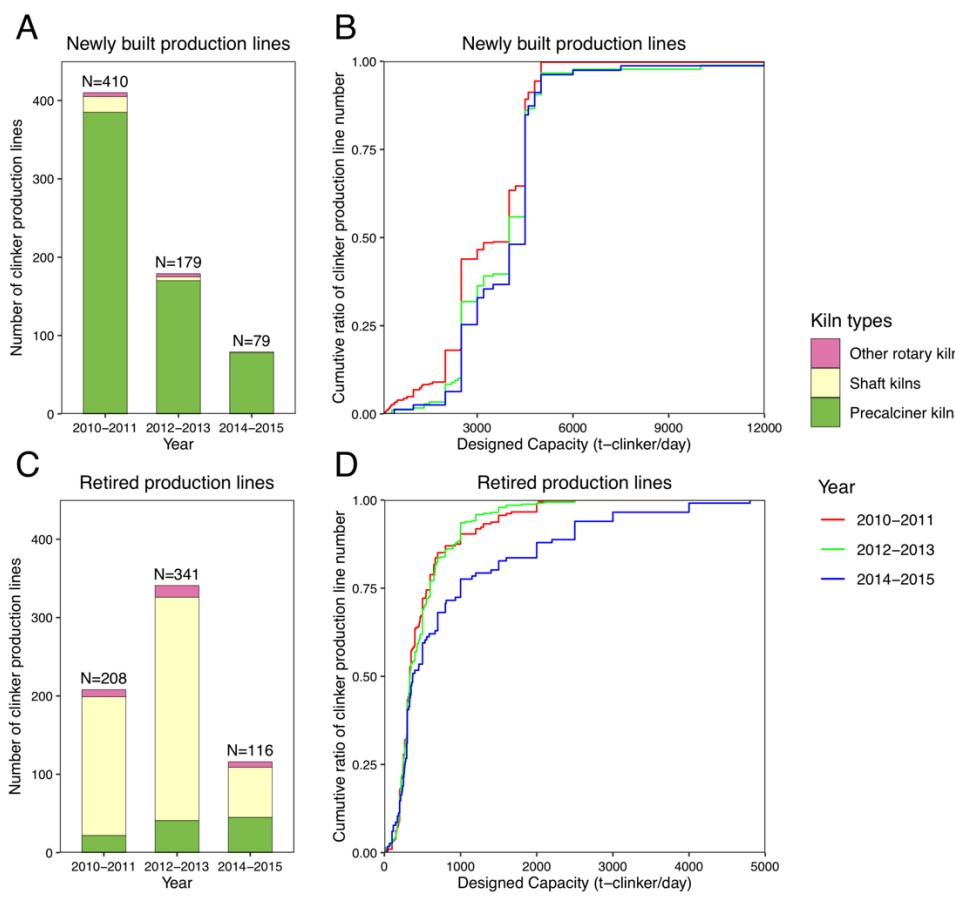


**Figure 4: Share of kiln types in newly built and retired production lines and cumulative ratio of unit number by capacity of the production lines.**



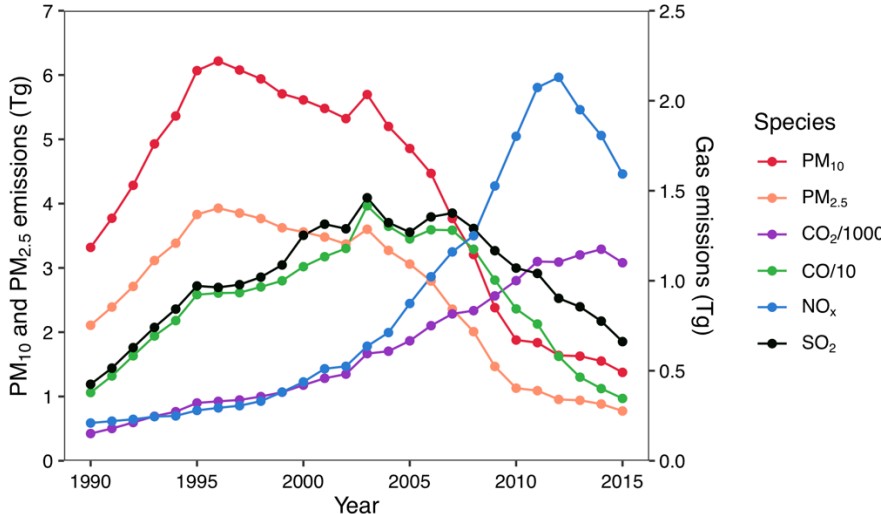


**Figure 5: Emissions of $SO_2$, $NO_x$, CO, $CO_2$, $PM_{2.5}$ and $PM_{10}$ in China's cement industry from 1990 to 2015.**





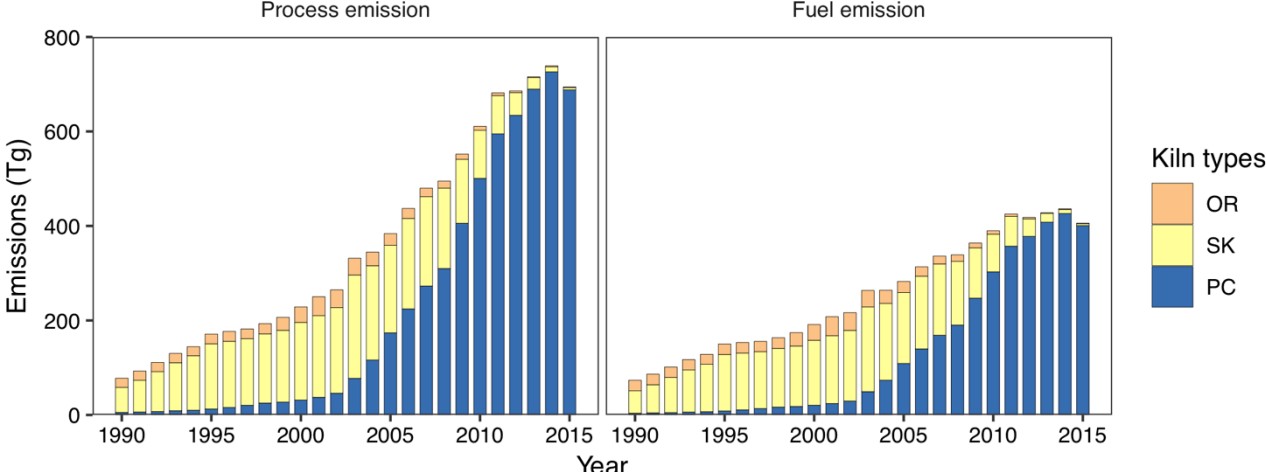

Figure 6: Historical CO$_2$ process and fuel emissions in China's cement industry from 1990 to 2015. The kiln types include the precalciner kilns (PC), shaft kilns (SK), and other rotary kilns (OR).





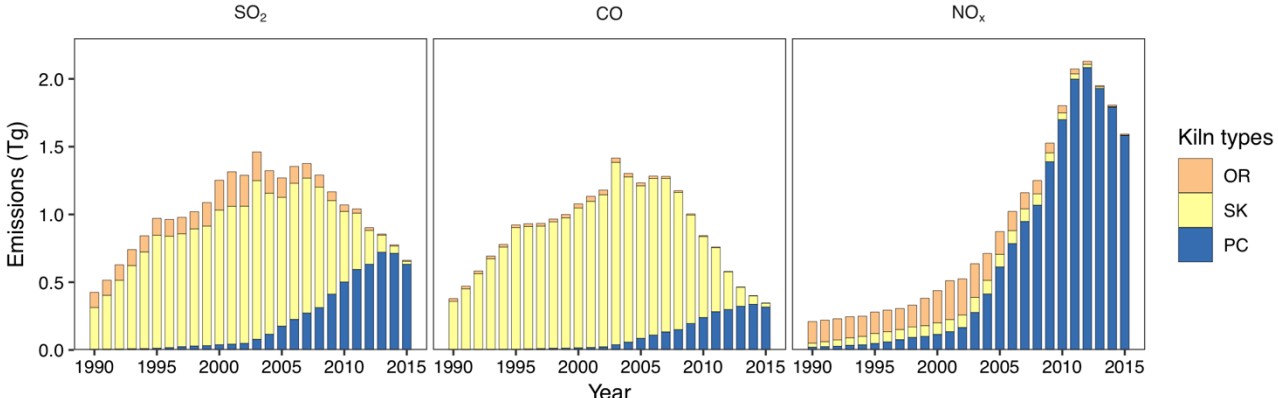

**Figure 7: Historical SO₂, CO, and NOₓ emissions by different kilns types from 1990 to 2015. The kiln types include the precalciner kilns (PC), shaft kilns (SK), and other rotary kilns (OR).**





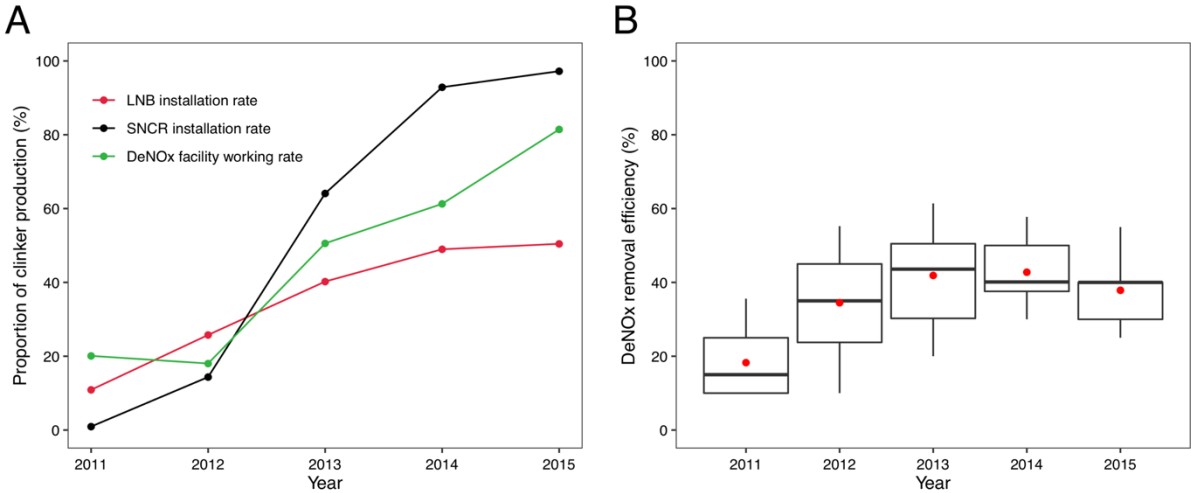


**Figure 8: The application proportion (of clinker production amount) of DeNO$_x$ technologies (LNB, SNCR) (A) and the average DeNO$_x$ removal efficiency of kilns in which the DeNO$_x$ facilities are working (B).**



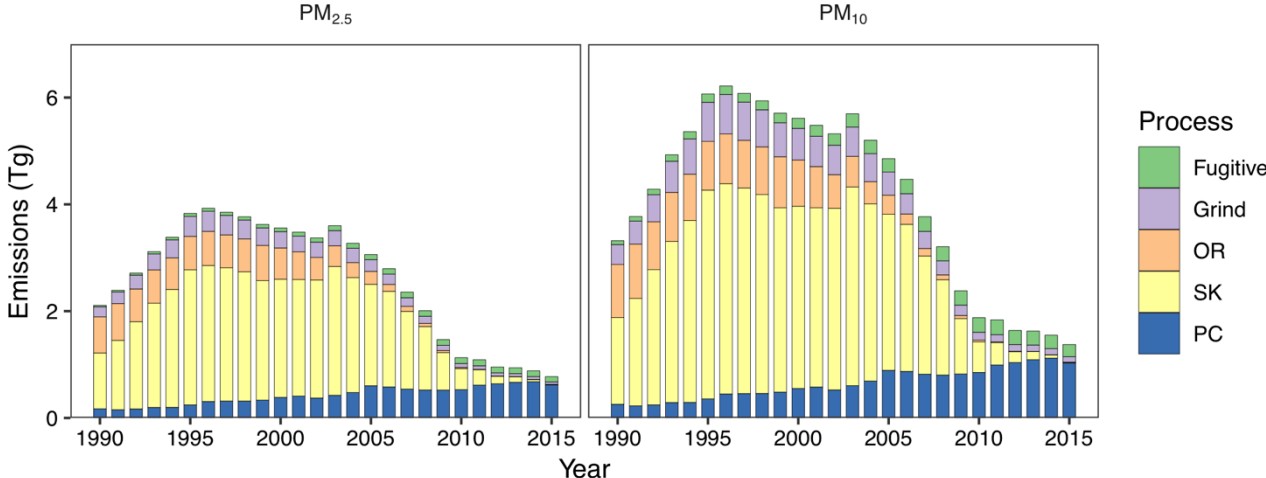


Figure 9: Historical PM$_{2.5}$ and PM$_{10}$ emissions by different processes from 1990 to 2015. The kiln types include the precalciner kilns (PC), shaft kilns (SK), and other rotary kilns (OR).





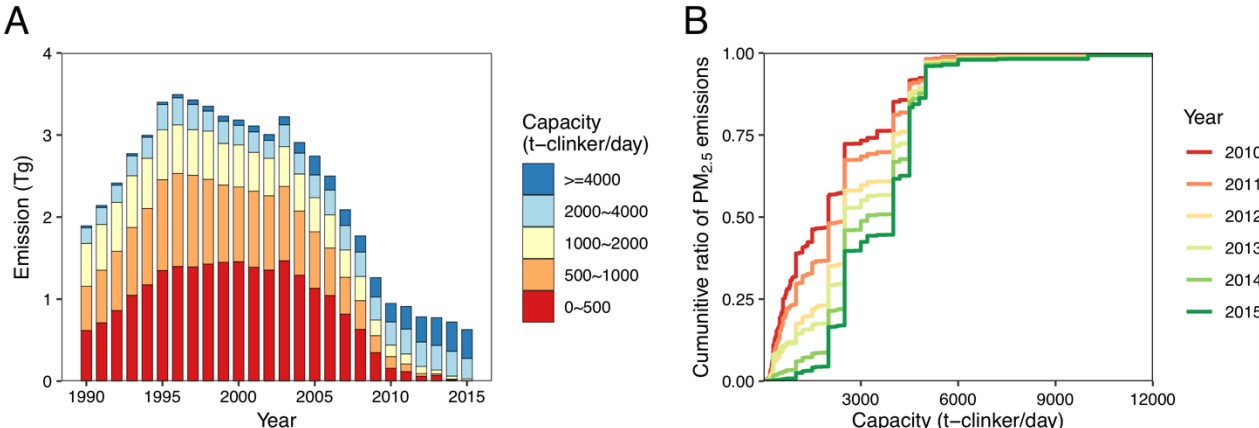


**Figure 10: Historical PM$_{2.5}$ emissions from the clinker calcination process by capacity (A) and cumulative ratio of PM$_{2.5}$ emissions by capacity of the production lines during the 2010-2015 period (B).**






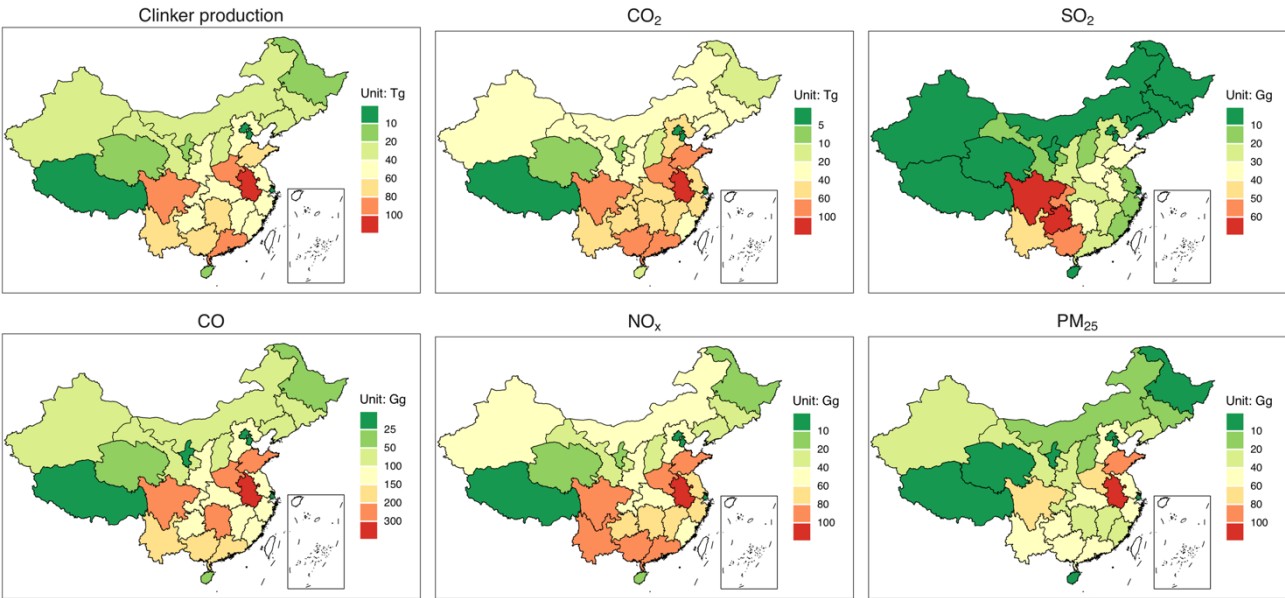

**Figure 11: Provincial clinker production and CO₂, SO₂, CO, NOₓ, and PM₂.₅ emissions from China's cement industry in 2015.**





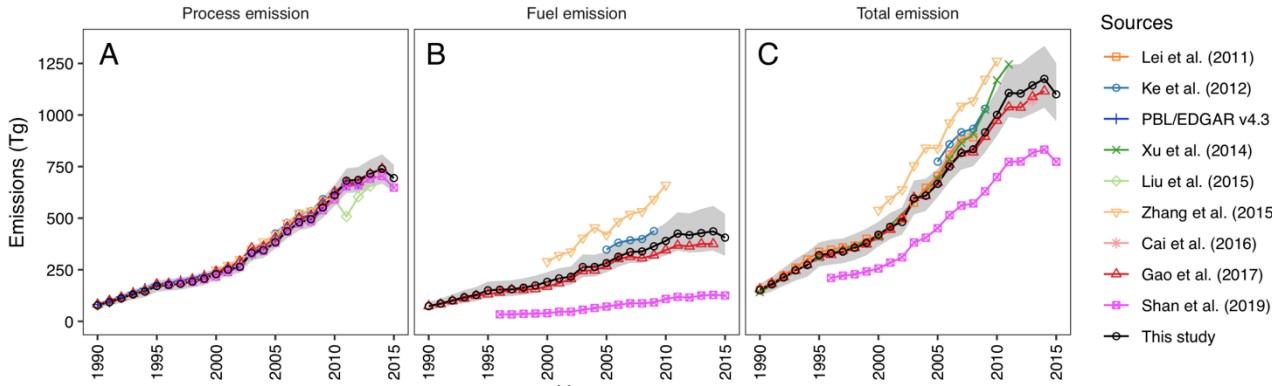


**Figure 12: Comparisons of CO₂ process emissions (A), fuel emissions (B), and total emissions (C) from China's cement industry during the 1990-2015 period. The gray shading illustrates the 95% confidence interval of the emission estimates in this study.**





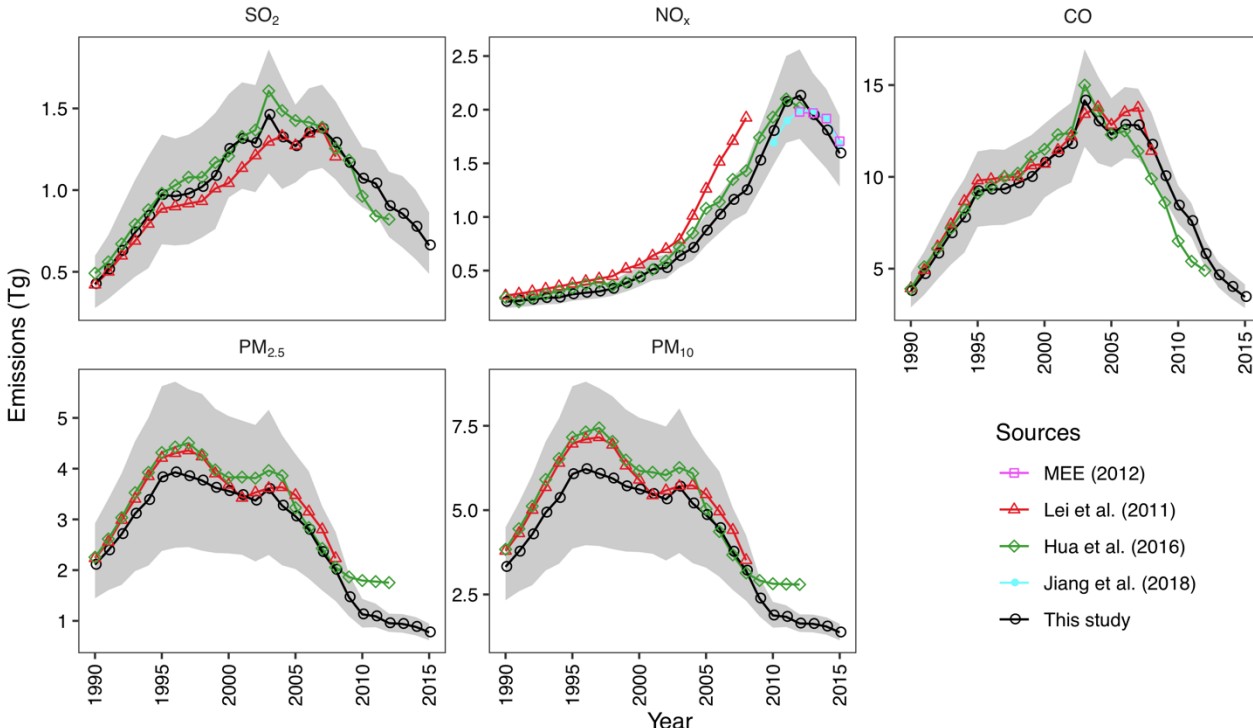


**Figure 13: Comparisons of SO₂, NOₓ, CO, PM₂.₅ and PM₁₀ emissions from China's cement industry during the 1990-2015 period. The gray shading illustrates the 95% confidence interval of the emission estimates in this study.**



**Table 1 Equations used for estimating emissions in China's cement industry**

| Pollutant | Equation for emission estimation |
|---|---|
| PM | $$E_{PM} = \sum_i P_{clinker,i} \times EF_{clinker,PM,i} \times (1-\eta_{clinker,i}) + \sum_i P_{cement,i} \times EF_{grind,PM,i} \times (1-\eta_{grind,i})$$ $$+ \sum_i P_{clinker,i} \times EF_{clinker,fugitive,PM,i} \times (1-\eta_{clinker,fugitive,i})$$ $$+ \sum_i P_{cement,i} \times EF_{grind,fugitive\,PM,i} \times (1-\eta_{grind,fugitive,\,i})$$ |
| NOx SO$_2$ CO | $$E_{gas} = \sum_i P_{clinker,i} \times EF_{clinker,gas} \times (1-\eta_i)$$ $$= \sum_i P_{clinker,i} \times EF_{coal,gas} \times EI_{clinker} \times (1-\eta_i)$$ |
| CO$_2$ | $$E_{CO_2} = \sum_i P_{clinker,i} \times EF_{calcinlatin,CO_2} + M_{coal,i} \times EF_{coal,CO_2}$$ |

*i*: the ID number of the cement production lines and grinding stations; *E*: the total emissions, *tons/year*; $P_{clinker}$: clinker
production, *tons/year*; $P_{cement}$: cement production, *tons/year*; $EF_{clinker,\,PM}$: organized PM emission factor during the clinker
calcination process, *g/kg*; $\eta_{clinker}$: removal efficiency PM control technology during the clinker calcination process;
$EF_{grind,\,PM}$: organized PM emission factor during the cement grinding process, *g/kg*; $\eta_{grind}$: removal efficiency PM control
technology during the cement grinding process; $EF_{clinker,\,fugitive,\,PM}$: fugitive PM emission factor during the clinker
calcination process, *g/kg*; $\eta_{clinker,\,fugitive}$: removal efficiency fugitive PM control technology during the clinker calcination
process; $E_{Fgrind,\,fugitive,\,PM}$: fugitive PM emission factor during the cement grinding process, *g/kg*; $\eta_{grind,\,fugitive}$: removal
efficiency of fugitive PM control technology during the cement grinding process; $EF_{clinker,\,gas}$: emission factor of gaseous
species (SO$_2$, NO$_x$, and CO) per ton of clinker produced, *g/kg*; $\eta$: removal efficiency of control technology for gaseous
species (particularly for NO$_x$); $EF_{coal,\,gas}$: emission factor of gaseous species per ton of coal consumed, *g/kg*; $EI_{clinker}$:
energy intensity of the clinker calcination process, *kg coal/kg clinker*; $EF_{calcination,\,CO2}$: CO$_2$ emission factor from clinker
calcination, *g/kg clinker*; $M_{coal}$: coal consumption during the clinker calcination process, tons/year; $EF_{coal,\,CO2}$: CO$_2$
emission factor from coal combustion, *g/kg coal*.





**Table 2 Emission factors of SO₂, NOx, CO, and CO₂ from cement kilns. The kiln types include precalciner kilns (PC), shaft kilns**
**(SK) and the other rotary kilns (OR).**

| Kiln types | $SO_2$[a,b] | $NO_x$[a] | CO[a] | $CO_2$ | Reference |
|---|---|---|---|---|---|
| PC | 3.2 | 10.9 | 15.35 | 519.66 g kg⁻¹ (clinker)<br>1940 g kg⁻¹ (coal) | Wang et al. 2008 |
| SK | 13.1 | 1.2 | 145.55 | 499.83 g kg⁻¹ (clinker)<br>1940 g kg⁻¹ (coal) | CRAES 2011<br>Lei et al. 2011<br>Shen et al. 2014 |
| OR | 11.4 | 13.8 | 17.8 | 499.83 g kg⁻¹ (clinker)<br>1940 g kg⁻¹ (coal) | Hua et al. 2016 |

[a.] unit: g/kg of coal combusted in the cement kilns
[b.] National average $SO_2$ emission factors weighted by coal consumption.



**Table 3 PM emission factors for clinker production, cement grinding, and fugitive emissions. The kiln types include precalciner**
**kilns (PC), shaft kilns (SK) and the other rotary kilns (OR).**

| Emission process | | Total PM | $PM_{2.5}$ | $PM_{2.5-10}$ | $PM_{>10}$ | EF ranges | References |
|---|---|---|---|---|---|---|---|
| Clinker production (g/kg clinker) | PC | 251.0 | 33.8 | 55.1 | 162.1 | 223.3~278.6 | Lei et al. (2011); Hua et al. (2016); CRAES 2011; |
| | SK | 129.5 | 14.2 | 26.9 | 88.4 | 88.7~170.4 | |
| | OR | 270.5 | 30.8 | 55.5 | 184.2 | 262.5~278.5 | |
| Cement grinding (g/kg cement) | | 35.1 | 1.4 | 4.2 | 29.5 | 20.3~50 | |
| Fugitive (g/kg product) | PC (≥4000 t clinker/day) | 0.2 | 0.02 | 0.04 | 0.14 | 0.1~0.3 | CRAES 2011; |
| | PC (2000~4000 t clinker/day) | 0.3 | 0.03 | 0.06 | 0.21 | 0.1~0.5 | |
| | PC (<2000 t clinker/day) | 0.45 | 0.045 | 0.09 | 0.315 | 0.15~0.75 | |
| | SK | 1.2 | 0.12 | 0.24 | 0.84 | 0.4~2.0 | |
| | OR | 1.2 | 0.12 | 0.24 | 0.84 | 0.4~2.0 | |
| | Grinding (≥0.6 million tons/year) | 0.6 | 0.06 | 0.12 | 0.42 | 0.2~1.0 | |
| | Grinding (<0.6 million tons/year) | 0.9 | 0.09 | 0.18 | 0.63 | 0.3~1.5 | |






**Table 4 Removal efficiencies of PM control technologies (%)**

| Technology | $PM_{25}$ | $PM_{2.5-10}$ | $PM_{>10}$ |
|---|---|---|---|
| Cyclone (CYC) | 10 | 70 | 90 |
| Wet scrubber (WET) | 50 | 90 | 99 |
| Electrostatic precipitator (ESP) | 93 | 98 | 99.5 |
| High-efficiency electrostatic precipitator (ESP2) | 96 | 99 | 99.9 |
| Bag filters (BAG) | 99 | 99.5 | 99.9 |



**Table 5 Cement production, capacity sizes, energy intensity, and clinker to cement ratio in China during 1990-**
**2015. The kiln types include precalciner kilns (PC), shaft kilns (SK) and the other rotary kilns (OR).**

| Category | Subcategory | 1990 | 1995 | 2000 | 2005 | 2010 | 2011 | 2012 | 2013 | 2014 | 2015 |
|---|---|---|---|---|---|---|---|---|---|---|---|
| Cement Production (Million tons/year) | PC | 14.0 | 34.0 | 79.6 | 473.7 | 1487.9 | 1800.4 | 1967.3 | 2350.8 | 2447.4 | 2337.8 |
| | SK | 143.2 | 384.6 | 431.3 | 525.2 | 367.5 | 280.8 | 230.1 | 63.2 | 38.3 | 16.2 |
| | OR | 52.6 | 57.1 | 86.1 | 69.9 | 26.6 | 18.0 | 12.5 | 5.2 | 6.4 | 5.4 |
| Capacity Size (%) | <2000 t-clinker/day | 87.6 | 88.8 | 86.0 | 59.3 | 24.4 | 18.7 | 12.5 | 7.4 | 4.6 | 2.7 |
| | 2000-4000 t-clinker/day | 10.5 | 9.8 | 10.5 | 23.4 | 29.1 | 29.9 | 30.3 | 30.7 | 30.4 | 28.5 |
| | >=4000 t-clinker/day | 1.9 | 1.5 | 3.4 | 17.3 | 46.5 | 51.4 | 57.3 | 61.9 | 65.0 | 68.8 |
| Energy Intensity (MJ/kg-clinker) | PC | 3.93 | 3.78 | 3.65 | 3.51 | 3.39 | 3.36 | 3.34 | 3.31 | 3.29 | 3.26 |
| | SK | 4.82 | 4.66 | 4.51 | 4.36 | 4.21 | 4.18 | 4.16 | 4.13 | 4.10 | 4.07 |
| | OR | 6.21 | 5.84 | 5.48 | 5.15 | 4.84 | 4.78 | 4.73 | 4.67 | 4.61 | 4.55 |
| Clinker to cement ratio (%) | | 74.0 | 71.8 | 76.2 | 70.6 | 62.9 | 62.8 | 59.9 | 57.0 | 57.1 | 56.6 |



**Table 6 Technology penetration, emission factors and emissions of the cement industry in China during the 1990-2015 period.**

| Category | Subcategory | 1990 | 1995 | 2000 | 2005 | 2010 | 2011 | 2012 | 2013 | 2014 | 2015 |
|---|---|---|---|---|---|---|---|---|---|---|---|
| Technology penetration (% of total clinker production) | LNB | 0.0 | 0.1 | 0.2 | 1.4 | 7.1 | 10.9 | 25.8 | 40.2 | 49.0 | 50.4 |
| | SNCR | 0.0 | 0.0 | 0.0 | 0.0 | 0.6 | 0.9 | 14.4 | 64.1 | 92.9 | 97.2 |
| | CYC | 46.2 | 41.5 | 26.5 | 12.8 | 0.5 | 0.3 | 0.1 | 0.1 | 0.1 | 0.0 |
| | WET | 45.9 | 44.2 | 30.1 | 14.7 | 3.3 | 2.2 | 1.1 | 0.8 | 0.3 | 0.1 |
| | ESP | 5.2 | 10.9 | 24.6 | 18.0 | 0.5 | 0.2 | 0.1 | 0.1 | 0.0 | 0.0 |
| | ESP2 | 0.0 | 0.0 | 4.2 | 17.2 | 21.2 | 19.9 | 18.5 | 16.3 | 14.7 | 13.0 |
| | BAG | 2.7 | 3.4 | 14.6 | 37.4 | 74.5 | 77.4 | 80.2 | 82.8 | 85.0 | 87.0 |
| Emission factor | $SO_2$ (g/kg cement) | 2.03 | 2.04 | 2.10 | 1.19 | 0.57 | 0.50 | 0.41 | 0.35 | 0.31 | 0.28 |
| | $NO_x$ (g/kg cement) | 1.00 | 0.59 | 0.73 | 0.82 | 0.96 | 0.99 | 0.96 | 0.81 | 0.72 | 0.68 |
| | CO (g/kg cement) | 18.07 | 19.40 | 18.06 | 11.53 | 4.48 | 3.62 | 2.62 | 1.92 | 1.61 | 1.47 |
| | $CO_2$ (kg/kg cement) | 0.72 | 0.68 | 0.70 | 0.62 | 0.53 | 0.53 | 0.50 | 0.47 | 0.47 | 0.47 |
| | $PM_{2.5}$ (g/kg cement) | 10.05 | 8.05 | 5.96 | 2.86 | 0.60 | 0.52 | 0.43 | 0.39 | 0.35 | 0.33 |
| | $PM_{10}$ (g/kg cement) | 15.83 | 12.76 | 9.40 | 4.54 | 1.00 | 0.88 | 0.74 | 0.67 | 0.62 | 0.58 |
| Emissions | $SO_2$ (Tg/year) | 0.43 | 0.97 | 1.25 | 1.27 | 1.07 | 1.04 | 0.90 | 0.86 | 0.78 | 0.66 |
| | $NO_x$ (Tg/year) | 0.21 | 0.28 | 0.44 | 0.87 | 1.80 | 2.07 | 2.13 | 1.95 | 1.81 | 1.59 |
| | CO (Tg/year) | 3.79 | 9.23 | 10.78 | 12.33 | 8.44 | 7.60 | 5.80 | 4.64 | 4.01 | 3.46 |
| | $CO_2$ (Pg/year) | 0.15 | 0.32 | 0.42 | 0.67 | 1.00 | 1.11 | 1.10 | 1.14 | 1.18 | 1.10 |
| | $PM_{2.5}$ (Tg/year) | 2.11 | 3.83 | 3.56 | 3.06 | 1.13 | 1.09 | 0.95 | 0.94 | 0.88 | 0.77 |
| | $PM_{10}$ (Tg/year) | 3.32 | 6.07 | 5.61 | 4.86 | 1.88 | 1.84 | 1.64 | 1.63 | 1.55 | 1.37 |
