# Peer review of "Carbon and air pollutant emissions from China's cement industry 1990-2015: trends, evolution of technologies and drivers"

_Atmospheric Chemistry and Physics, 2020_

## Referee Comment (RC1) · Anonymous Referee #1 · 1 Nov 2020

Cement industry is one of the largest contributors to the industrial emissions of carbon dioxide and air pollutants in China. Based on intensive unit-based information, this study investigated the carbon and air pollutant emissions from China's cement industry during 1990-2015, explored the emission trends, evolution of technologies and drivers to changes of emissions. This work contributed to the development of China's high-resolution emission inventory, which is very useful for the atmospheric community. The manuscript also provided new insights for future emission mitigation of China's cement industry. The topic is within the scope of ACP and the manuscript is generally well written. I have a few comments before it can be accepted for publication.

[Figure]

One major advantage of the new emission inventory is the unit level data. I believe this should be emphasized throughout the manuscript.

Is the cement output in 2014 the same with the output in 2015? Otherwise the growth rate between 1990-2014 and 1990-2015 should be different. Page 1, Line 16: "We found that, from 1990 to 2015, accompanied by a 10.9-fold increase in cement production, $CO_2$ , $SO_2$, and $NO_x$ emissions from China's cement industry increased by 626%, 59%, and 658%, whereas CO, $PM_{2.5}$ and $PM_{10}$ emissions decreased by 9%, 66%, and 63%, respectively. " Page 8, Line 246: "From 1990 to 2014, the production of cement and clinker increased from 0.21 and 0.16 billion tons to 2.5 and 1.4 billion tons, i.e., by 10.9 and 8.2 times, respectively."

Page10,Line 282- 291,in the chapter of 3.2.1 $CO_2$ emissions,the contents are mixing the period of 1990-2014 with the period of 1990-2015, which is unclear to readers.

Page 11,Line 320-323,"The decline of PM emissions after 1996 was due to the implementation of the new emission standards for the cement industry issued in 1996 (GB4915-1996, Table S1) and the slowing down of the economy in the Asian financial crisis. The PM emissions rebounded after the financial crisis but dropped again after 2003, despite a continuous increase in cement production at an annual growth rate higher than 10%." In Fig. 9, the $PM_{2.5}$ emissions kept decreasing during 1990-2002, and only rebounded in 2003. It's difficult to judge whether the rebound is due to the financial crisis or not.

There is some inconsistency of the numbers. The authors should carefully double check the data: (1) Page 1, Line 16,"We found that, from 1990 to 2015, accompanied by a 10.9-fold increase in cement production, $CO_2$ , $SO_2$, and $NO_x$ emissions from China's cement industry increased by 626%, 59%, and 658%, whereas CO, $PM_{2.5}$ and $PM_{10}$ emissions decreased by 9%, 66%, and 63%, respectively. " Page 9, Line 275,During the 25 years, the cement production increased dramatically, by 10.5

times. During that time, the CO2 , SO2 , and NOx emissions from the cement industry increased by 627%, 56%, and 659%, whereas the CO, PM2.5 and PM10 emissions decreased by 9%, 63%, and 59%, respectively, indicating that significant technology transitions occurred in the past 25 years. Page 15, Line,438ïijŇ"From 1990 to 2015, the CO2 , SO2 , and NOx emissions from the cement industry increased by 627%, 56%, and 659%, whereas the CO, PM2.5 and PM10 emissions decreased by 9%, 63%, and 59%, respectively. " (2) Page 6ïijŇLine 169- 291ïijŇ"From 2011 to 2015, the proportion of kilns equipped with LNB technology increased from 3% to 40%, and the installation percentage of LNB in newly established kilns increased from 13% to 64%. The SNCR technology developed later in the 2000s. During the 12th FYP, the SNCR installation experienced unprecedented explosive growth. The penetration rate has increased even faster than that of the LNB technology, from 1% of all the kilns in service in 2011 to 88% in 2015. " Page10ïijŇLine 307-308ïijŇ"In 2011, only 11% and 1% of the clinker was manufactured in kilns equipped with LNB and SNCR facilities, whereas by 2015, the percentages sharply increased to 50% and 97%." Please explain the meaning of the cumulative ratio occurred in Fig. 4 and Fig. 10 in more details.

---

## Referee Comment (RC2) · Anonymous Referee #3 · 5 Nov 2020

This study inventories the emissions from China's cement industry. The manuscript is clear and well written. However, I have the following concerns that should be addressed before considering publishing.

1. The unit-level data of 2010-2015 is more interesting. Please show more results and analysis at the unit-level.

2. The unit-level of clinker and cement production for the years 1990-2009 are scaled based on data of 2010, thus lead to huge uncertainties. Is there any grey literature to show the changes in the national/provincial production of clinker and cement that could be used to adjust calibrate the extrapolated parameters?

[Figure]

3. Page 4 line 105-107. Why do you use linear regression to eliminate the differences between different studies? Then you assume there is a linear relationship between energy intensity and time, which is not true. We usually use mean value or median value instead.

4. The title says "drivers", but I do not see any driving analysis of cement production and related emissions. The whole study is based on inventory accounting. Quantifying the drivers are very important for the reduction solutions. What drivers caused the increased in cement production and related emissions, and what drivers caused the decline in CO, PM2.5 and PM10 and by how much per cent? We all know that production technology innovation could reduce emissions from the cement industry, but the question is how good are their effects? By how much per cent can production technology innovation reduce the emissions?

5. To me, the major contribution of this study is inventorying emissions from cement plants. Thus, I urge the authors to consider publish their data with this manuscript for wider academic use and policymaking. Although the raw unit-level data are owned by the Ministry of Ecology and Environment, which are confidential, it is still possible to share your calculated emission data with the academic society.

---

## Author Comment (AC1) · 1 Dec 2020

Cement industry is one of the largest contributors to the industrial emissions of carbon dioxide and air pollutants in China. Based on intensive unit-based information, this study investigated the carbon and air pollutant emissions from China's cement industry during 1990-2015, explored the emission trends, evolution of technologies and drivers to changes of emissions. This work contributed to the development of China's high resolution emission inventory, which is very useful for the atmospheric community. The manuscript also provided new insights for future emission mitigation of China's cement industry. The topic is within the scope of ACP and the manuscript is generally well written. I have a few comments before it can be accepted for publication.

**Response:** We thank the Referee for the insightful comments. We have revised the manuscript according to the suggestions and respond to the concerns below.

1. One major advantage of the new emission inventory is the unit level data. I believe this should be emphasized throughout the manuscript.

**Response:** Accepted. Thanks to the review's comment, we have rewritten some of the contents in the manuscript and added unit-level emission analysis to emphasize the advantage of new unit-level emission inventory:

**(1) Abstract**: "*In 2010, nationwide 39% and 31% of the $PM_{2.5}$ and $NO_x$ emission were produced by 3% and 15% of the total capacity of the production lines, indicating the disproportionate high emissions from a small number of the super-polluting units*".

**(2) Introduction**: "*Based on the background above, the aim of this study is to quantify the decadal changes of carbon dioxide and air pollutant emissions from China's cement industry, investigate the evolution technologies, identifying the super-polluting units, and quantify the major drivers of the emission changes over a period of 25 years. The analysis is based on intensive unit-based information on activity rates, production capacity, operation status, and control technologies, which improves the accuracy of the estimation of cement emissions, provides a comprehensive view of the effectiveness of technologies on air pollutant emission control in the past, quantifies the contribution from different drivers to de changes of emissions, and highlights the opportunities and challenges for future mitigation of carbon dioxide and air pollutant emissions in China.* "

**(3) Results (3.2.4 Unit-level emissions)**: "*Fig. 11 shows the unit-level $PM_{2.5}$ and $NO_x$ emissions by capacity in 2010 and 2015, which highlights the most polluting production lines whose emission intensity is over 90th percentile values of the emission intensity defined as the emissions per unit of capacity. During 2010−2015, dramatic changes had taken place in China's cement industry. In 2010, there were over 2400 cement production lines, in which PC had a share of 54% in terms of the number of production lines, followed by SK, with a considerable share of 44%. Typically, the SKs had smaller capacities and older ages, which were majorly within the range of 100−1000 t-clinker/day and started to operate before 2000, but had substantial contributions to $PM_{2.5}$ emissions. In 2010, nationwide 39% and 31% of the $PM_{2.5}$ and $NO_x$ emission were produced by 3% and 15% of the total capacity, indicating the dipropionate high emissions from a small number of the super-polluting units. Specifically, the super-polluting units for $PM_{2.5}$ were dominated by SKs, whereas the super-polluting units for $NO_x$ were majorly PCs. In 2015, driven by the rapid replacement of traditional SKs with PCs, and the elimination small-scale production lines, the disproportionalities were alleviated compared with the situation in 2015. Allowing for the dominant role of PC in China's cement industry since 2015, future mitigation should focus on the control of cement demand growth, improvement of energy efficiency, and implementation of high-efficiency end-of-pipe emission control devices.*

[Figure]

*Figure 11 in manuscript: Unit-level $PM_{2.5}$ and $NO_x$ emissions during clinker calcination in production lines by capacity in 2010 and 2015. The black lines and gray shades illustrate the production lines whose emission intensity is over 90th percentile values of the emission intensity defined as the emissions per unit of capacity.*
"

2. Is the cement output in 2014 the same with the output in 2015? Otherwise the growth rate between 1990-2014 and 1990-2015 should be different. Page 1, Line 16: "We found that, from 1990 to 2015, accompanied by a 10.9-fold increase in cement production, $CO_2$ , $SO_2$, and $NO_x$ emissions from China's cement industry increased by 626%, 59%, and 658%, whereas CO, $PM_{2.5}$ and $PM_{10}$ emissions decreased by 9%, 66%, and 63%, respectively. " Page 8, Line 246: "From 1990 to 2014, the production of cement and clinker increased from 0.21 and 0.16 billion tons to 2.5 and 1.4 billion tons, i.e., by 10.9 and 8.2 times, respectively.

**Response:** Accepted. We've made a calculation mistake here. The cement production increase from 0.21 billion tons in 1990 to 2.49 billion tons in 2014, and then dropped to 2.36 billion tons in 2015. Therefore, the cement growth rate between 1990 and 2014 is 10.9 times, and the growth rate between 1990 and 2015 is 10.3 times. We've corrected the numbers in the manuscript. Besides, we've also

corrected the inconsistent numbers as mentioned in comment No. 5.

**(1) Page 1, Line 16**: "*We found that, from 1990 to 2015, accompanied by a 10.3-fold increase in cement production, $CO_2$, $SO_2$, and $NO_x$ emissions from China's cement industry increased by 627%, 56%, and 659%, whereas CO, $PM_{2.5}$ and $PM_{10}$ emissions decreased by 9%, 63%, and 59%, respectively.*"

**(2) Page 8, Line 246**: "*From 1990 to 2014, the production of cement and clinker increased from 0.21 and 0.16 billion tons to 2.49 and 1.42 billion tons, i.e., by 10.9 and 8.2 times, respectively.*"

3. Page10: Line 282- 291, the chapter of 3.2.1 $CO_2$ emissions, the contents are mixing the period of 1990-2014 with the period of 1990-2015, which is unclear to readers.

**Response:** Accepted. We have rewritten the paragraph to make the contents consistent:

[revised manuscript text omitted]

**Response:** Accepted. We have carefully double-checked the data, and corrected the inconsistent

numbers.

**(1)** Page 1, Line 16: "*We found that, from 1990 to 2015, accompanied by a 10.3-fold increase in cement production, $CO_2$, $SO_2$, and $NO_x$ emissions from China's cement industry increased by 627%, 56%, and 659%, whereas CO, $PM_{2.5}$ and $PM_{10}$ emissions decreased by 9%, 63%, and 59%, respectively.*"

Page 9, Line 275: "*During the 25 years, the cement production increased dramatically, by 10.3 times. During that time, the $CO_2$ , $SO_2$ , and $NO_x$ emissions from the cement industry increased by 627%, 56%, and 659%, whereas the CO, $PM_{2.5}$ and $PM_{10}$ emissions decreased by 9%, 63%, and 59%, respectively, indicating that significant technology transitions occurred in the past 25 years.*"

Page 15, Line 438: "*From 1990 to 2015, the $CO_2$, $SO_2$, and $NO_x$ emissions from the cement industry increased by 627%, 56%, and 659%, whereas the CO, $PM_{2.5}$ and $PM_{10}$ emissions decreased by 9%, 63%, and 59%, respectively.*"

**(2)** The inconsistency is caused by differences in description. Previously we mix the proportion in the number of kilns with the proportion in the amount of clinker produced in the kilns. We've clarified them in in the revised manuscript.

Page 6, Line 169- 291: "*From 2011 to 2015, the proportion in the number of kilns equipped with LNB technology increased from 5% to 40%, and correspondingly, the proportion of clinker manufactured in kilns equipped with LNB facility increased from 11% to 50%. The installation percentage of LNB in newly established kilns increased from 13% to 64%. The SNCR technology developed later in the 2000s. During the 12$^{th}$ FYP, the SNCR installation experienced unprecedented explosive growth. The penetration rate has increased even faster than that of the LNB technology, from 1% of the number of kilns in service in 2011 to 88% in 2015, and thus the proportion of clinker manufactured in kilns equipped with SNCR facility increased from 1% to 97%.*"

Page10, Line 307-308, "*In 2011, only 11% and 1% of the clinker was manufactured in kilns equipped with LNB and SNCR facilities, whereas by 2015, the percentages sharply increased to 50% and 97%.*"

6. Please explain the meaning of the cumulative ratio occurred in Fig. 4 and Fig. 10 in more details.

**Response:** Accepted. We explained the meaning of cumulative ratio in Section 3.1, near the occurrence of Fig. 4:

"*To draw the curve for the cumulative ratio, we summarized the number of production lines by capacity (t-clinker/day), and calculated the ratio to the total number of production lines, from which we derived*

*the cumulative ratio for each level of capacity. Therefore, the cumulative ratio represents the share of production lines with the capacity below a certain level.*"

---

## Author Comment (AC2) · 1 Dec 2020

This study inventories the emissions from China's cement industry. The manuscript is clear and well written. However, I have the following concerns that should be addressed before considering publishing.

**Response:** We appreciate the Referee's helpful comments. Below we have point-by-point addressed the Referee's concerns.

1. The unit-level data of 2010-2015 is more interesting. Please show more results and analysis at the unit-level.

**Response:** Accepted. We have added more unit-level results by presenting the relationship between capacity and annual emissions of $PM_{2.5}$ and $NO_x$ from cement production lines, and discussed the future mitigation directions in section **3.2.4 Unit-level emissions**:

" *Fig. 11 shows the unit-level $PM_{2.5}$ and $NO_x$ emissions by capacity in 2010 and 2015, which highlights the most polluting production lines whose emission intensity is over 90th percentile values of the emission intensity defined as the emissions per unit of capacity. During 2010−2015, dramatic changes had taken place in China's cement industry. In 2010, there were over 2400 cement production lines, in which PC had a share of 54% in terms of the number of production lines, followed by SK, with a considerable share of 44%. Typically, the SKs had smaller capacities and older ages, which were majorly within the range of 100−1000 t-clinker/day and started to operate before 2000, but had substantial contributions to $PM_{2.5}$ emissions. In 2010, nationwide 39% and 31% of the $PM_{2.5}$ and $NO_x$ emission were produced by 3% and 15% of the total capacity, indicating the dipropionate high emissions from a small number of the super-polluting units. Specifically, the super-polluting units for $PM_{2.5}$ were dominated by SKs, whereas the super-polluting units for $NO_x$ were majorly PCs. In 2015, driven by the rapid replacement of traditional SKs with PCs, and the elimination small-scale production lines, the disproportionalities were alleviated compared with the situation in 2015. Allowing for the dominant role of PC in China's cement industry since 2015, future mitigation should focus on the control of cement demand growth, improvement of energy efficiency, and implementation of high-efficiency end-of-pipe emission control devices.*

[Figure]

*Figure 11 in manuscript: Unit-level PM$_{2.5}$ and NO$_x$ emissions during clinker calcination in production lines by capacity in 2010 and 2015. The black lines and gray shades illustrate the production lines whose emission intensity is over 90th percentile values of the emission intensity defined as the emissions per unit of capacity.*
"

2. The unit-level of clinker and cement production for the years 1990-2009 are scaled based on data of 2010, thus lead to huge uncertainties. Is there any grey literature to show the changes in the national/provincial production of clinker and cement that could be used to adjust calibrate the extrapolated parameters?

**Response:** Accepted. In order to make the unit-level data for the years of 1990-2009 as realistic as possible, we combined all the available data from the MEE database, statistics and literature to build the clinker and cement output for each cement production line. Specifically, we first calculated the provincial clinker and cement output from the data sources mentioned above, and then distributed the provincial amount among the cement production lines in each province for each year by considering the capacity, kiln type, age of each production line. Therefore, in the emission database, the data on national and provincial clinker and cement output are consistent with existing data from statistics and literature. Whereas the data on unit-level clinker and cement production have higher uncertainties,

since they are derived based on the information of the capacity, kiln type, age of each production line. To address the reviewer's concern, we've revised the contents in section **2.1 Activity rates** with more details on the methodology of developing the unit-level of clinker and cement production for the years of 1990-2009:

" *Based on the MEE database for 2010-2015, we derived the unit-level activity rates for the period 1990-2009, with a combination of data from statistics and literature. We first calculated the provincial clinker and cement output from the existing data sources, and then distributed the yearly provincial output among the cement production lines in each province by considering the age, kiln type and capacity of each production line. In details, we obtained the national and provincial cement output during 1990-2009 from China Statistical Yearbook (National Bureau of Statistics, 1991-2010a) and China Industry Economy Statistical Yearbook (National Bureau of Statistics, 1991-2010b), and collected the national (2002-2009) and provincial (2005-2009) clinker output from China Cement Almanac(China Cement Association, 2001-2010). Additional data on provincial clinker output for some discrete years (such as 1993, 1994 and 1997) before 2005 were obtained from China Industry Economy Statistical Yearbook (National Bureau of Statistics, 1991-2010b). The data on national clinker to cement ratio during 1990-2001 were adopted from literature (Xu et al., 2012, 2014; Gao et al., 2017). To derive the clinker output for the early years, on national scale, we calculated the clinker output as the product of clinker to cement ratio and the cement output for years of 1990-2001. On provincial scale, we derived the clinker to cement ratio for each year of 1990-2004 based on a linear interpolation with the available year-specific provincial clinker to cement ratio from statistics, and calculated the provincial clinker output as the product of provincial clinker to cement ratio and the provincial cement output, using the national clinker output as a constrain. Therefore, in the emission database, the data on national and provincial clinker and cement output are consistent with existing data from statistics and literature, but unit-level activity prior to 2010 are more uncertain because it is extrapolated based on the information of the age, kiln type and capacity of each production line.* "

3. Page 4 line 105-107. Why do you use linear regression to eliminate the differences between different studies? Then you assume there is a linear relationship between energy intensity and time, which is not true. We usually use mean value or median value instead.

**Response:** Owing to the replacement of outdated kilns with advanced kilns, and the implementation

of energy efficiency measures, the energy intensity of cement industry decreased with time, which was stated in many previous studies (Lei et al., 2011; Xu et al., 2014; Gao et al., 2017). If we use the mean or median values, the trend of energy intensity will be perturbed by extreme numbers reported in the literature (black circles in the Figure 1). Besides, we need to extrapolate the coal use intensity during 1990-2012 to the period of 2013-2015, and the rapid decrease of the coal use intensity in other rotary kilns (OR) in 2012 is questionable. Therefore, we choose to derive the coal use intensity through regression.

[Figure]

Figure 1 the coal use intensity for each year derived from the numbers from literature with median and mean values.

To address the reviewer's concern on the linear regression, we tried a non-linear regression with Generalized Additive Model (GAM) as a sensitivity test. We added the discussions on the sensitivity test in the **Supplement** and added it as one source of uncertainties and limitation in section **5 Conclusions**:

**(1) Supplement:** "*Besides the linear model, we tried the non-linear regression with Generalized Additive Model (GAM) as a sensitivity test. GAM is a semi-parametric approach which can predict non-linear responses to selected predictor variables. As shown in Figure 2, we compared the regression of the logarithm of coal use intensity for different kiln types with linear and non-linear method. We found that the GAM regression of the logarithm of coal use intensity has slight higher r square in the regressions for PC and OR, and predicts shaper decrease of coal use intensity in recent years. However, the 95% confidential intervals of both curves were overlapping, illustrating no*

*significant differences between the two types of regressions.*

[Figure]

*Figure S1 in Supporting Information: Regression of the logarithm of coal use intensity for different kiln types with linear and non-linear (GAM) method. The shadings illustrates the 95% confidential interval of the regression curves. The kiln types include precalciner kilns (PC), shaft kilns (SK) and the other rotary kilns (OR). Further, we compared the emission results of $CO_2$, $SO_2$, $NO_x$, and CO, which were estimated through the coal-use based emission factors. Fig. S2 shows the emission ratio between the emission estimates through the coal use derived by GAM regression and the emission estimates through the coal use derived by the linear regression for $CO_2$, CO, $SO_2$, and $NO_x$. The GAM regression predicted higher emission estimates during 1995-2007, and lower emission estimates during 1990-1994 and 2008-2015. The relative differences between both estimates were within the ranges of ±5%, which were much lower than the overall uncertainty ranges of the emission estimates. Therefore, considering the simple explicit expression, we present the final results with the coal use intensity predicted by the linear regression model.*

[Figure]

*Figure S2 in Supporting Information: Emission ratio between the emission estimates through the coal use*

*derived by GAM regression and the emission estimates through the coal use derived by the linear regression for CO₂, CO, NOₓ, and SO₂."*

**(2) Conclusions:** " *We predicted the coal use intensity by the linear regression between the logarithm of energy intensity and time in years, which may underestimate the improvement in the energy efficiency of clinker production in recent years. Unit-based coal use data is helpful in narrowing the gaps between model estimation and the real world situation.* "

4. The title says "drivers", but I do not see any driving analysis of cement production and related emissions. The whole study is based on inventory accounting. Quantifying the drivers are very important for the reduction solutions. What drivers caused the increased in cement production and related emissions, and what drivers caused the decline in CO, PM2.5 and PM10 and by how much per cent? We all know that production technology innovation could reduce emissions from the cement industry, but the question is how good are their effects? By how much per cent can production technology innovation reduce the emissions?

**Response:** Accepted. We added the contents of the driver analysis as follow:

**(1) 2.3 Drivers to changes of emissions:** "*We made a unit-level quantification of the contributions from six factors to the net changes of CO₂ and air pollutant emissions, i.e., cement production, changes of kiln types, improvement of energy efficiency, reduction of clinker to cement ratio, reduction of sulphur content in coal, and implementation of the end-of-pipe control measures. Following our previous study on the power sector (Liu et al., 2015; Wu et al., 2019), for a given period, we developed a series of hypothetical scenarios to estimate the contribution from each factor incrementally. For example, for the period of 2010-2015, we built the baseline scenario by changing the cement output from the amount in 2010 to the amount in 2015, and then changed the other five factors incrementally to the situation in 2015. The difference between every consecutive step is an estimate of the contribution of each factor. Since the order of the factors may change the results, we calculated the average factor contributions through all the change sequences in the factors. We applied the method of hypothetical scenarios rather than the index decomposition approaches (such the logarithmic mean divisia index, LMDI) since we hope explicitly quantify the effects of drivers at unit level.*

**(2) 3.4 Drivers to changes of emissions:** " *The trends in SO₂, NOₓ, PM₂.₅, and CO₂ emissions are affected by a variety of factors. As shown in Fig. 13, the growth of cement production continuously*

*contributed to the increase of CO₂ and air pollutant emissions. The evolution of cement production technology from the shaft kilns to precalciner kilns has led to the dramatic decrease of SO₂ emissions, but contributed to the increase of NOₓ and PM₂.₅ emissions, since the precalciner kilns have higher NOₓ and PM₂.₅ emission factors than the shaft kilns. The decrese of energy intensity would decrease the coal use demand per unit cement output, and the reduction of clinker to cement ratio would result in lower demand of coal and lime stone, which both contributed to a continuous decrease of air pollutant and CO₂ emissions. The reduction of sulphur content in coal was helpful in reducing SO₂ emissions. Prominently, the end-of-pipe control measures were the major driver to the remarkable decline of PM and NOₓ emissions. Overall, however, the SO₂, NOₓ and CO₂ emissions were still 56%, 659%, and 627% higher than the levels in 1990. Further steps including implementation of energy efficiency measures and promotion of high-efficiency SO₂ and NOₓ removal technologies are crucially needed to effectively reduce the emissions from the cement industry.*

[Figure]

*Figure 13 in manuscript: Contribution of factors to the national emission changes of SO₂, NOₓ, PM₂.₅ and CO₂ during 1990-2015.*"

5. To me, the major contribution of this study is inventorying emissions from cement plants. Thus, I urge the authors to consider publish their data with this manuscript for wider academic use and policymaking. Although the raw unit-level data are owned by the Ministry of Ecology and

Environment, which are confidential, it is still possible to share your calculated emission data with the academic society.

**Response:** Accepted. We've published all the data in the figures of the manuscript, including the unit-level emissions in Figure 11 in figshare (https://doi.org/10.6084/m9.figshare.c.5223113.v1). The high-resolution cement emission inventory has been incorporated into the MEIC model (http://www.meicmodel.org/), which is available to the community.

**References**

China Cement Association: China Cement Almanac, China building industry press, Beijing., 2001-2010.

Gao, T., Shen, L., Shen, M., Liu, L., Chen, F. and Gao, L.: Evolution and projection of $CO_2$ emissions for China's cement industry from 1980 to 2020, Renew. Sust. Energ. Rev., 74, 522–537, doi:10.1016/j.rser.2017.02.006, 2017.

Lei, Y., Zhang, Q., Nielsen, C. and He, K.: An inventory of primary air pollutants and $CO_2$ emissions from cement production in China, 1990–2020, Atmos. Environ., 45(1), 147–154, doi:10.1016/j.atmosenv.2010.09.034, 2011.

Liu, F., Zhang, Q., Tong, D., Zheng, B., Li, M., Huo, H. and He, K. B.: High-resolution inventory of technologies, activities, and emissions of coal-fired power plants in China from 1990 to 2010, Atmos. Chem. Phys., 15(23), 13299–13317, doi:10.5194/acp-15-13299-2015, 2015.

National Bureau of Statistics: China Statistical Yearbook, China Statistics Press, Beijing., 1991-2010a.

National Bureau of Statistics: China Industry Economy Statistical Yearbook, China Statistics Press, Beijing., 1991-2010b.

Wu, R., Liu, F., Tong, D., Zheng, Y., Lei, Y., Hong, C., Li, M., Liu, J., Zheng, B., Bo, Y., Chen, X., Li, X. and Zhang, Q.: Air quality and health benefits of China's emission control policies on coal-fired power plants during 2005–2020, Environ. Res. Lett., 14(9), 094016, doi:10.1088/1748-9326/ab3bae, 2019.

Xu, J.-H., Fleiter, T., Eichhammer, W. and Fan, Y.: Energy consumption and $CO_2$ emissions in China's cement industry: A perspective from LMDI decomposition analysis, Energy Policy, 50, 821–832, doi:10.1016/j.enpol.2012.08.038, 2012.

Xu, J.-H., Fleiter, T., Fan, Y. and Eichhammer, W.: $CO_2$ emissions reduction potential in China's cement industry compared to IEA's Cement Technology Roadmap up to 2050, Appl. Energy, 130, 592–602, doi:10.1016/j.apenergy.2014.03.004, 2014.